# Stable isotopes of nitrate reveal different nitrogen processing mechanisms in streams across a land use gradient during wet and dry periods

Wei Wen Wong[1], Jesse Pottage[1], Fiona Y. Warry[1], Paul Reich[1.2], Keryn L. Roberts[1], Michael R. Grace[1], Perran L.M. Cook[1]

[1]Water Studies Centre, School of Chemistry, Monash University, Clayton, 3800, Australia
[2]Arthur Rylah Institute for Environmental Research, Department of Environment, Land Water and Planning, Heidelberg, 3084, Australia

*Correspondence to*: Wei Wen Wong (weiwen.wong@monash.edu)

**Abstract.** Understanding the relationship between land use and the dynamics of nitrate ($NO_3^-$) is the key to constrain sources of $NO_3^-$ export in order to aid effective management of waterways. In this study, isotopic compositions of $NO_3^-$ ($\delta^{15}N$-$NO_3^-$ and $\delta^{18}O$-$NO_3^-$) were used to elucidate the effects of land use (agriculture in particular) and rainfall on the major sources and sinks of $NO_3^-$ within the Westernport catchment, Victoria, Australia. This study is one of the very few studies carried out in temperate regions with highly stochastic rainfall patterns; enabling a more comprehensive understanding of the applications of $NO_3^-$ isotopes in catchment ecosystems with different climatic conditions. Longitudinal samples were collected from five streams with different agriculture land use intensities on five occasions – three during dry periods and two during wet periods. At the catchment scale, we observed significant positive relationships between $NO_3^-$ concentrations ($p < 0.05$), $\delta^{15}N$-$NO_3^-$ ($p < 0.01$) and percentage agriculture (particularly during the wet period) reflecting the dominance of anthropogenic nitrogen inputs within the catchment. Different rainfall conditions appeared to be major controls on the predominance of the sources and transformation processes of $NO_3^-$ in our study sites. Artificial fertiliser was the dominant source of $NO_3^-$ during the wet periods. In addition to artificial fertiliser, nitrified organic matter in sediment was also an apparent source of $NO_3^-$ to the surface water during the dry periods. Denitrification was prevalent during the wet periods while uptake of $NO_3^-$ by plants or algae was only observed during the dry periods in two streams. The outcome of this study suggests that effective reduction of $NO_3^-$ load to the streams can only be achieved by prioritising management strategies based on different rainfall conditions.

## 1 Introduction

Anthropogenic sources of $NO_3^-$ from catchments can pose substantial risk to the quality of freshwater ecosystems (Vitousek et al. 1997; Galloway et al. 2004; Galloway et al. 2005). Over-enrichment of $NO_3^-$ in freshwater systems is a major factor in development of algal blooms which often promote bottom water hypoxia and anoxia. Such anoxia intensifies nutrient recycling and can lead to disruption of ecosystem functioning and ultimately loss of biodiversity (Galloway et al. 2004; Carmago and Alonso 2006). Freshwater streams are often sites for enhanced denitrification (Peterson et al. 2001; Barnes and Raymond

2010). However, when $NO_3^-$ loading from the catchment exceeds the removal and retention capacity of the streams, $NO_3^-$ is transported to downstream receiving waters including estuaries and coastal embayments, which are often nitrogen-limited, further compounding the problem of eutrophication.

Understanding the sources, transport and sinks of $NO_3^-$ is critical, particularly in planning and setting guidelines for better management of the waterways (Xue et al. 2009). Establishing the link between land use and the biogeochemistry of $NO_3^-$ provides fundamental information to help develop $NO_3^-$ reduction and watershed restoration strategies (Kaushal et al. 2011). To date, the most promising tool to investigate the sources and sinks of $NO_3^-$ are the dual isotopic compositions of $NO_3^-$ at natural abundance level (expressed as $\delta^{15}N$-$NO_3^-$ and $\delta^{18}O$-$NO_3^-$ in ‰). Preferential utilisation of lighter isotopes ($^{14}N$ and $^{16}O$) over heavier isotopes ($^{15}N$ and $^{18}O$) leads to distinctive isotopic signatures that differentiate the various $NO_3^-$ sources/end members (e.g. inorganic and organic fertiliser, animal manure, atmospheric deposition) and the predictable kinetic fractionation effect when $NO_3^-$ undergoes different biological processes (e.g. nitrogen fixation and denitrification). For instance, numerous previous culture-based experiments revealed that denitrification and phytoplankton assimilation fractionate N and O isotopes equally (1:1 pattern) leaving behind $NO_3^-$ that is enriched in both $^{15}N$ and $^{18}O$ (Fry 2006). Simultaneous measurement of $\delta^{15}N$-$NO_3^-$ and $\delta^{18}O$-$NO_3^-$ also provides complementary information on the cycling of $NO_3^-$ in the environment. $\delta^{18}O$-$NO_3^-$ is a more effective proxy of internal cycling of $NO_3^-$ (i.e. assimilation, mineralisation and nitrification) compared to $\delta^{15}N$-$NO_3^-$. This is because during $NO_3^-$ assimilation and mineralisation, N atoms are recycled between fixed N pools and the O atoms are removed and replaced by nitrification (Sigman et al. 2009; Buchwald et al. 2012).

In addition to constraining $NO_3^-$ budget and N cycling in various environmental settings, previous studies have also utilized the dual isotopic signatures of $NO_3^-$ to study the effects of different land uses on the pool of $NO_3^-$ in headwater streams (Barnes and Raymond 2010, Sebilo et al. 2003), creeks (Danielescu and MacQuarrie 2013) and large rivers (Voss et al. 2006; Battaglin et al. 2001). Barnes and Raymond (2010) for example found that both $\delta^{15}N$-$NO_3^-$ and $\delta^{18}O$-$NO_3^-$ varied significantly between urban, agricultural and forested areas in the Connecticut River watershed, USA. Several other investigators (Mueller et al. 2016; Mayer et al. 2002) showed positive relationships between $\delta^{15}N$-$NO_3^-$ and the percent of agricultural land in their study area, indicating the applicability of $\delta^{15}N$-$NO_3^-$ and $\delta^{18}O$-$NO_3^-$ to distinguish $NO_3^-$ originating from different land uses. Danielescu and MacQuarrie (2013) and Chang et al. (2002) on the other hand, found no correlations between $NO_3^-$ isotopes and land use intensities in the Trout River catchment and the Mississippi River Basin; respectively. These studies attributed the lack of correlation to catchment size (Danielescu and MacQuarrie, 2013) and the homogeneity of land use (Chang et al. 2002).

Despite the extensive application of $NO_3^-$ isotopes to study the transport of terrestrial $NO_3^-$ to the tributaries in the catchment; majority of these studies were carried out in the United States and Western Europe where climatic conditions, for example temperature and rainfall patterns are different compared to that in the southern hemisphere. The southern hemisphere tends to have more sporadic and variable rainfall patterns compared to the northern hemisphere and Australia is an example of this. The variable rainfall patterns can modulate different efficiencies of denitrification in soils and thus different fractionation effects to the residual $NO_3^-$ pool (Chien et al. 1977, Billy et al. 2010). However, the lack of $NO_3^-$ isotope studies

in the southern hemisphere (Ohte et al. 2013) impedes a more thorough understanding of $NO_3^-$ dynamics within catchment ecosystems.

Most previous studies investigating the relationship between land use and $NO_3^-$ export using $\delta^{15}N$-$NO_3^-$ and $\delta^{18}O$-$NO_3^-$ have either focused on the seasonal or spatial variations in one stream, or used multiple streams with one site per stream (i.e. Mayer et al. 2002; Yevenes et al. 2016). Far fewer studies have incorporated longitudinal sampling of multiple streams over multiple seasons. Nitrate concentrations and concomitant isotopic signatures can change substantially, not only spatially but temporally. Changes in hydrological and physicochemical (notably temperature) conditions of a river can affect the relative contribution of different sources of $NO_3^-$ and the seasonal predominance of a specific source (Kaushal et al. 2011; Panno et al. 2008). In some studies (e.g. Riha et al. 2014; Kaushal et al. 2011), denitrification and assimilation by plants and algae have been reported to be more prominent during the dry seasons compared to the wet seasons but in other studies (e.g. Murdiyarso et al. 2010; Enanga et al. 2016) denitrification appeared to be more prevalent during the wet seasons as precipitation induces saturation of soils resulting in oxygen depletion and thereby low redox potentials that favour denitrification. As such, if spatial and temporal variations of $\delta^{15}N$-$NO_3^-$ and $\delta^{18}O$-$NO_3^-$ are not considered thoroughly in a sampling regime, it can lead to misinterpretation of the origin and fate of $NO_3^-$. Proper consideration of the temporal variability of $NO_3^-$ isotope signatures and transformation are particularly pertinent in catchments with highly stochastic rainfall patterns, such as Australia.

In this study, we examine both spatial and temporal variations of $NO_3^-$ concentrations and isotopic compositions within and between 5 streams in 5 catchments spanning an agricultural land-use gradient, enabling us to evaluate (1) the effects of agriculture land use on the sources and transformation processes of $NO_3^-$ and (2) the effects of rainfall on the predominance of the sources and fate of $NO_3^-$ in the catchments.

## 2 Materials and methods

### 2.1 Study area

This study was undertaken using 5 major streams (Bass River, Lang Lang River, Bunyip River, Watsons Creek and Toomuc Creek) draining into Western Port (Fig. 1) which lies approximately 75km south east of Melbourne, Australia. Western Port is a nitrogen-limited coastal embayment (CSIRO, 1996) recognised as a Ramsar site for migratory birds. The catchments in the Western Port contain three marine national parks, highlighting its environmental and ecological significance. The catchments cover an area of 3,721 $km^2$ with land uses ranging from semi-pristine/state forest to high density residential and intense agricultural activities. The area experiences a temperate climate with average annual rainfall ranging from 750mm along the coast to 1200mm in the northern highlands. Mean monthly rainfall was about 20mm and 53mm in 2014 and 2015, respectively (Australian Bureau of Meteorology 2014 - http://www.bom.gov.au/).

The catchment overlies a multi-layered combined aquifer system. The main aquifer consists of Quaternary alluvial and dune deposit (average thickness of <7m) as well as Baxter, Sherwood and Yallock formations (average thickness between

20 and 175m). These aquifers are generally unconfined with radial groundwater flow direction from the basin edge towards Western Port bay. The hydrogeology of Western Port can be found in Carillo-Rivera, 1975.

Five longitudinal surveys were carried out between April 2014 and May 2015, two during wet periods (14/4/2014; 15/5/2015 - the total rainfall for 5 days before sampling was between 45 and 65mm) and three during dry periods (8/4/2014; 22/5/2014; 21/3/2015 - the total rainfall for 5 to 10 days before sampling was <5mm). A total of 21 sampling sites, indicated in Fig. 1 were selected across a gradient of catchment land use intensity. The five streams were selected based on the extent and distribution of land use types between and within each stream sub-catchment (see Fig. S1 in supplementary material), thus enabling comparisons within and between the streams.

In this study, catchment intensive agriculture was used as predictor of land use intensity in the catchment. These data were obtained from the National Environmental Stream Attributes database v1.1 (Stein et al. 2014), Bureau of Rural Sciences' 2005/06 Land Use of Australia V4 maps (www.agriculture.gov.au/abares/aclump) and Victorian Resources Online (VRO). In the context of this study, the catchment intensive agriculture variable is termed as 'percentage agriculture'. This term represents the percentage of the catchment subject to intensive animal production, intensive plant production (horticulture and irrigated cropping) and grazing of modified pastures. This variable also reflects the integrated diffuse sources of nutrients derived from intense agriculture including animal manure and inorganic fertilisers. The percentage agriculture for the sampling sites ranged between 2 to 96% with the Bass River (94±2%) > Lang Lang (79±5%) > Watsons (76±4%) > Toomuc (71±16%) > Bunyip (upper Bunyip: 12±9%; lower Bunyip: 54±10%; Fig. 2). For the purpose of this study, Bunyip is divided into two sectors (upper and lower Bunyip) based on the proximity of the sampling sites (Fig. 1) and the percentage of land use. All the sampling sites in the upper Bunyip are situated in areas with >30% forestation (see Fig. S1). In general, the percentage agriculture decreases with increased distance from the Western Port Bay (WPB) for all the streams except Bass River. There is an increase of about 2% in percentage agriculture for Bass River with increased distance from WPB. Watsons Creek has the largest percentage of market gardens (~91%).

## 2.2 Sample collection and preservation

Water quality parameters (pH, electrical conductivity, turbidity, dissolved oxygen (DO) concentration and water temperature) were measured using a calibrated Horiba U-10 multimeter. Stream samples were collected for the analyses of dissolved inorganic nutrients-DIN (ammonium, $NH_4^+$; $NO_3^-$ and nitrite, $NO_2^-$), dissolved organic carbon (DOC) and $NO_3^-$ isotopes ($\delta^{15}N$-$NO_3^-$ and $\delta^{18}O$-$NO_3^-$). These samples were filtered on site using 0.2µm Pall Supor® membrane disc filters. Filtered DOC samples were acidified to pH < 2 with concentrated hydrochloric acid. Samples for $\delta^{18}O$-$H_2O$ were collected directly from the streams without filtering. Sediment samples were collected from the bottom of the rivers and were kept in zip-lock bags. All samples were stored and transported on ice until they were refrigerated (nutrients samples were frozen) in the laboratory. In addition to stream water and sediment, we also collected four samples of artificial/inorganic fertiliser (from the fertiliser distributor in the area) and five cow manure (from local farmers).

## 2.3 DIN and DOC concentration measurements

All chemical analyses were performed within 1-2 weeks of sample collection except for isotope analyses (within 2 months). The concentrations of $NO_3^-$, $NO_2^-$, and $NH_4^+$ were determined spectrophotometrically using a Lachat QuikChem 8000 Flow Injection Analyzer (FIA) following standard procedures (APHA 2005). DOC concentrations were determined using a Shimadzu TOC-5000 Total Organic Carbon analyser. Analysis of standard reference materials indicated the accuracy of the spectrophotometric analyses and the TOC analyser was always within 2% relative error.

## 2.4 Isotopic analyses

The samples for $\delta^{15}N$-$NO_3^-$ and $\delta^{18}O$-$NO_3^-$ were analysed using the chemical azide method based on the procedure outlined in McIlvin et al. (2005). In brief, $NO_x$ ($NO_3^-$ + $NO_2^-$) was quantitatively converted to $NO_2^-$ using cadmium reduction and then to $N_2O$ using sodium azide. The initial $NO_2^-$ concentrations were insignificant, typically <1% relative to $NO_3^-$. Hence, the influence of $\delta^{15}N$–$NO_2^-$ was negligible and the measured $\delta^{15}N$-$N_2O$ represents the signature of $\delta^{15}N$-$NO_3^-$. The resultant $N_2O$ was then analysed on a Hydra 20-22 continuous flow isotope ratio mass spectrometer (CF-IRMS; Sercon Ltd., UK) interfaced to a cryoprep system (Sercon Ltd., UK). Nitrogen and oxygen isotope ratios are reported in per mil (‰) relative to atmospheric air (AIR) and Vienna Standard Mean Ocean Water (VSMOW), respectively. The external reproducibility of the isotopic analyses lies within ± 0.5‰ for $\delta^{15}N$ and ± 0.3‰ for $\delta^{18}O$. The international reference materials used were USGS32, USGS 34, USGS 35 and IAEA-$NO_3^-$. Lab-internal standards ($KNO_3^-$ and $NaNO_2^-$) with pre-determined isotopic values were also processed the same way as the samples to check on the efficiency of the analytical method. The $\delta^{18}O$-$H_2O$ values were measured via equilibration with He-$CO_2$ at 32°C for 24 to 48 hours in a Finnigan MAT Gas Bench and then analysed using CF-IRMS. The $\delta^{18}O$-$H_2O$ values were referenced to internal laboratory standards, which were calibrated using VSMOW and Standard Light Antarctic Precipitation. Measurement of two sets of triplicate samples in every run showed a precision of 0.2‰ for $\delta^{18}O$-$H_2O$. Sediment samples for the analysis of $\delta^{15}N$ of total nitrogen were dried at 60°C before being analysed on the 20-22 CF-IRMS coupled to an elemental analyzer (Sercon Ltd. UK). The precision of the elemental analysis and $\delta^{15}N$ was 0.5µg and ±0.2‰ (n=5), respectively.

## 2.5 Data Analysis

The relationships between percentage agriculture and surface water $NO_3^-$ concentrations were assessed using linear regression. Percentage agriculture was the predictor variable. $NO_3^-$ concentration, and $\delta^{15}N$-$NO_3^-$ were response variables. Relationships between $\delta^{15}N$-$NO_3^-$ and $NO_3^-$ concentration as well as $\delta^{18}O$-$NO_3^-$ and $\delta^{15}N$-$NO_3^-$ were assessed using Pearson's correlation. The $NO_3^-$ isotopes response variables were assessed at two spatial scales – individual stream and catchment scale. The catchment scale integrates data from all five studied streams. Any graphical patterns or relationships derived from using these scales represent processes that occur somewhere in the catchment either in the streams or prior to entering the streams with data from the individual stream is likely to represent more localised processes to that particular stream.

## 3 Results

The streams were oxic throughout the course of our study period with %DO saturation between 60 and 110% (see Fig. S2 in supplementary material). There was no apparent spatial and temporal variation in DO; however, %DO saturation was slightly lower during the dry periods (average of 73±20%) compared to the wet periods (average of 82±12%). Temperature was also relatively consistent with an average of 13±2°C. Ammonium concentration was generally low (<4 µM) except during the wet periods in Bunyip (~7 µM), Lang Lang (~21 µM) and Bass (~29 µM). DOC concentrations were typically 0.8±0.4 mM. Nitrite concentrations were also low in all the streams; ranged between 0.1µmol/L and 0.4µmol/L.

The spatial and temporal variations of $NO_3^-$ concentration, $\delta^{15}N$ and $\delta^{18}O$ across the sites are shown in Fig. 3. $NO_3^-$ concentrations ranged between 7 µM and 790 µM with averages of 21±15 µM, 50±130 µM, 64±43 µM, 71±43 µM and 190±280 µM for Toomuc, Bunyip, Bass, Lang Lang and Watsons, respectively. The lowest $NO_3^-$ concentration was observed in the lower Bunyip (4 µM) while the highest $NO_3^-$ concentration was observed in Watsons Creek (790 µM) at the most downstream site. Nitrate concentrations were generally higher during the wet periods compared to the dry periods in all streams (Fig. 3). During the wet periods, $NO_3^-$ concentrations typically followed an increasing trend heading downstream except for the Bass River which exhibited the opposite $NO_3^-$ trend with lower concentrations at downstream sites. During the dry periods, only the Bunyip and Bass Rivers showed apparent longitudinal patterns in $NO_3^-$ concentrations; with decreasing concentrations moving downstream in both. Sites with high percentage agriculture generally also exhibited high $NO_3^-$ concentrations (Fig. 4), particularly during the wet periods.

Overall, $\delta^{15}N$ of the riverine $NO_3^-$ spanned a wide range (+4 to +33‰). Approximately 62% of the obtained $\delta^{15}N$-$NO_3^-$ values fell below +10‰. More enriched $\delta^{15}N$-$NO_3^-$ values (> +10‰) were typically observed during the dry periods and were coincident with a high percentage agriculture (Fig. 4). Among all sites, $\delta^{15}N$-$NO_3^-$ values in the Bunyip and Bass were relatively depleted (+4 to +12‰ for Bunyip and +10 to 12‰ for Bass), with the lower range found at upper Bunyip (+4 to +8‰). There was no discernible pattern spatially or temporally in $\delta^{18}O$-$NO_3^-$, except that higher values were found in Lang Lang and Bass during the wet periods with +4 to +6‰ and +5 to +9‰; respectively compared to the dry periods (<+4‰). For other sampling sites, $\delta^{18}O$-$NO_3^-$ ranged between +2 to +13‰. The isotope compositions of sediment, water, artificial fertiliser and cow manure/organic fertiliser are presented in Table 1. The $\delta^{15}N$-TN of three potential sources – artificial fertiliser, organic fertiliser and soil organic matter ranged from -0.5 to +0.7‰, +6 to +13‰ and +4 to +5‰, respectively.

## 4 Discussion

### 4.1 Potential sources of $NO_3^-$

There are three major potential sources of $NO_3^-$ in the catchments – artificial fertiliser, cow manure/organic fertiliser and soil organic matter (SOM) – see Table 1 for the $\delta^{15}N$-TN values. The average $\delta^{15}N$-TN value of soils is used to directly represent the soil organic portion as most of the nitrogen in soils is generally bound in organic forms. Nitrogen isotope of the

$NO_3^-$ produced from the potential end members usually retains the signature of the $\delta^{15}N$-TN as a result of tight coupling between mineralisation (production of ammonium from organic matter) and nitrification (oxidation of ammonium to $NO_3^-$). The $\delta^{18}O$ of $NO_3^-$ generated by nitrification of these sources ($\delta^{18}O$-$NO_3^-$ $_{final}$) is, however; decoupled from $\delta^{15}N$-$NO_3^-$. As shown in Equation (1) which is adapted from Buchwald et al. 2012, $\delta^{18}O$-$NO_3^-$ $_{final}$ relies on the oxygen isotope of water ($\delta^{18}O$-$H_2O$), oxygen isotope of dissolved oxygen ($\delta^{18}O$-$O_2$), the kinetic isotope fractionation associated with incorporation of oxygen during ammonia oxidation ($^{18}\varepsilon_k$-$O_2$), kinetic isotope fractionation associated with incorporation of oxygen from water during ammonia oxidation ($^{18}\varepsilon_k$-$H_2O,_1$) and nitrite oxidation ($^{18}\varepsilon_k$-$H_2O,_2$), equilibrium isotope effect associated with oxygen isotope exchange between nitrite and water ($^{18}\varepsilon_{eq}$) as well as the fraction of nitrite oxygen atoms exchanged with $H_2O$ during ammonia oxidation ($x_{AO}$) (Casciotti et al. 2010; Buchwald et al. 2012). To date, $^{18}\varepsilon_k$-$O_2$ and $^{18}\varepsilon_k$-$H_2O$ cannot be separated. Previous culture studies have reported the overall $^{18}\varepsilon_k$-$O_2$ + $^{18}\varepsilon_k$-$H_2O,_1$ to range between 17.9‰ to 37.6‰ (Casciotti et al. 2010) while $^{18}\varepsilon_k$ -$H_2O,_2$ ranged from 12.8‰ to 18.2‰ (Buchwald and Casciotti 2010). These values together with $^{18}\varepsilon_{eq}$ value of 14‰, average $\delta^{18}O$-$H_2O$ of -5.3‰ and $\delta^{18}O$-$O_2$ of 23.5‰ were used to calculate the maximum and minimum estimates of the $\delta^{18}O$ of newly produced $NO_3^-$ from nitrification. The minimum estimate of $\delta^{18}O$-$NO_3^-$ $_{final}$ was calculated using the lower range of $^{18}\varepsilon_k$-$O_2$ + $^{18}\varepsilon_k$-$H_2O,_1$ (17.9‰) and $^{18}\varepsilon_k$ -$H_2O,_2$ (12.8‰) while the maximum estimate was calculated using the upper range of $^{18}\varepsilon_k$-$O_2$ + $^{18}\varepsilon_k$-$H_2O,_1$ (37.6‰) and $^{18}\varepsilon_k$ -$H_2O,_2$ (18.2‰). Based on the assumptions that ammonia was fully oxidised to $NO_3$- (as no accumulation of $NO_2^-$ was observed during our study period) and there was complete exchange of oxygen isotope between nitrite and $H_2O$ during ammonia oxidation ($x_{AO}$=1), which likely characterizes most freshwater systems (Casciotti et al. 2007, Snider et al. 2010, Buchwald and Casciotti 2013); we calculated the $\delta^{18}O$ of produced $NO_3^-$ from nitrification to be between -2.03‰ and -0.23‰.

Equation 1:

$$\delta^{18}O_{NO_3^-,final} = \left[\frac{2}{3} + \frac{1}{3}x_{AO}\right]\delta^{18}O_{H_2O} + \frac{1}{3}\left[(\delta^{18}O_{O_2} - {}^{18}\varepsilon_{k,O_2} - {}^{18}\varepsilon_{k,H_2O,1})(1 - x_{AO}) - {}^{18}\varepsilon_{k,H_2O,2}\right] + \frac{2}{3}{}^{18}\varepsilon_{eq}x_{AO}$$

The $\delta^{15}N$-TN of cow manure (+6 to +13‰) was most variable compared to other end members. This variation reflects the extent of volatilisation, a highly fractionating process. Volatilisation can cause a fractionation effect of up to 25‰ in the residual $NH_4^+$ (Hubner 1986). As such, the lower value of +6‰ indicates a relatively fresh manure sample and is assumed to represent the initial $\delta^{15}N$ of the cow manure before undergoing any extensive fractionation.

Atmospheric deposition did not appear to be an important source of $NO_3^-$ in this study based on the relatively depleted $\delta^{18}O$-$NO_3^-$ values (ranged from +2 to +8‰ during the wet periods; +1.5 to +13‰ during the dry periods) of the riverine samples. The $\delta^{18}O$-$NO_3^-$ of atmospheric deposition were reported to range from +60 to +95‰ in the literature (Kendall 2007; Elliott et al. 2007; Pardo et al. 2004). Similarly, groundwater was not considered as an important source of $NO_3^-$ to the streams based on the low $NO_3^-$ concentrations (~0.7 to 7.0µM) reported in previous studies (Water Information System Online; http://data.water.vic.gov.au/monitoring.htm).

## 4.2 General characteristics of $NO_3^-$ in the streams

Agricultural land use (i.e. market gardens and cattle rearing) appeared to influence $NO_3^-$ concentrations in our study sites. As shown in Fig. 4(a), during the wet periods, high $NO_3^-$ concentrations (> 40 $\mu$M) were particularly observed at sites with more than 70% agricultural land use. During the dry periods, although $NO_3^-$ concentrations were generally lower than 36$\mu$M, the outliers were observed at sites with more than 70% agricultural land use. Similarly, enriched $\delta^{15}N$-$NO_3^-$ in the streams were mainly found at sites with high percentage agricultural land use (between 75 to 85%) for both dry and wet periods suggesting that enriched $\delta^{15}N$-$NO_3^-$ in the stream were originated from agricultural activities. In fact, the most enriched $\delta^{15}N$-$NO_3^-$ values (>30‰) were observed at the most downstream site of Watson Creek which has the largest percentage of market gardens (although the total agricultural area is not the highest amongst all the studied sites). We also observed a significant positive relationship between $\delta^{15}N$-$NO_3^-$ and percentage agriculture during the wet periods ($r^2$ = 0.39, p<0.01; Fig. 4b). This further supports the contention that agricultural activities were the main control of the $\delta^{15}N$-$NO_3^-$ in the streams. Other researchers (e.g. Mayer et al. 2002 and Voss et al. 2006) have also documented similar trends of enriched $\delta^{15}N$-$NO_3^-$ with increasing percentage agriculture. For example Harrington et al. 1998, Mayer et al. 2002 and Voss et al. 2006 observed highly significant positive relationships between percentage agriculture land area and $\delta^{15}N$-$NO_3^-$ with $r^2 \sim 0.7$. However, these studies showed comparatively narrower and more depleted ranges of $\delta^{15}N$-$NO_3^-$ with 2.0 to 7.3‰; 4 to 8‰ and -0.1 to 8.3‰; respectively, suggesting more subtle changes in $\delta^{15}N$-$NO_3^-$ over a large span of agriculture land areas in these studies compared to our study.

Given that none of the predicted sources of $NO_3^-$ in the Western Port catchment exhibited an initial $\delta^{15}N$-$NO_3^-$ of more than +6‰, the isotopically-enriched $NO_3^-$ as well as the variability of $NO_3^-$ concentrations observed in this study were consequences of a series of transformation processes. Hence, we propose the following factors to explain the heavy isotopes and the different $NO_3^-$ concentrations across different periods observed in our study:

(1) During the wet period when surface runoff was conspicuous and residence time of the water column was low, in-stream $NO_3^-$ comprised mainly of externally derived $NO_3^-$ (i.e. fertilisers, manure and soil organic matter) and there was limited in-stream processing of these $NO_3^-$. The high $NO_3^-$ concentrations and the heavy $\delta^{15}N$-$NO_3^-$ values reflect the occurrence of mineralisation, nitrification and subsequent preferential denitrification of the isotopically lighter $NO_3^-$ source/s in either the waterlogged soil or in the soil zone underneath the market gardens before transport to the streams through surface runoff.

(2) During the dry periods when surface runoff was negligible and residence time of the water column was high, there was minimal introduction of external $NO_3^-$ into the streams and in-stream processing of $NO_3^-$ was more apparent than during the wet periods. In addition to mineralisation and nitrification, volatilisation and assimilation by plant and algae was highly likely to occur in the stream further reducing the $NO_3^-$ concentration and further fractionating the isotopic signature of $NO_3^-$.

These processes are conceptualised in Fig. 5 and are corroborated in the following discussion using two graphical methods: the Keeling plot and the isotope biplot. In an agricultural watershed, the co-existence of multiple sources and transformation processes can potentially complicate the use of $NO_3^-$ isotopes as tracers of its origin. Keeling plots ($\delta^{15}N$-$NO_3^-$ versus $1/[NO_3^-]$) are generally very useful to distinguish between mixing and fractionation (i.e. assimilation and bacterial denitrification) processes (Kendall et al. 1998). The latter typically results in progressively increasing $\delta^{15}N$-$NO_3^-$ values as $NO_3^-$ concentrations decrease and yields a curved Keeling plot. Meanwhile, mixing of $NO_3^-$ from two or more sources can result in concomitant increase of both $\delta^{15}N$-$NO_3^-$ and $NO_3^-$ concentrations and results in a straight line on the Keeling plot (Kendall et al. 1998). A biplot ($\delta^{18}O$-$NO_3^-$ versus $\delta^{15}N$-$NO_3^-$) on the other hand, is a proven diagnostic method to elucidate the presence of two isotope fractionating processes; assimilation and denitrification.

## 4.3 Key controlling processes of nitrate during the wet periods

In-stream processing of $NO_3^-$ was not evident during the wet periods based on the lack of relationships between $\delta^{18}O$-$NO_3^-$ and $[NO_3^-]$ as well as between $\delta^{18}O$-$NO_3^-$ and $\delta^{15}N$-$NO_3^-$ for the individual streams (shown in Supplementary Fig. S3). If denitrification was dominant, both $\delta^{15}N$-$NO_3^-$ and $\delta^{18}O$-$NO_3^-$ values are expected to increase in a 1:1 pattern at low $NO_3^-$ concentration – a trend which has been proven by numerous culture-based experiments to indicate the occurrence of denitrification. (Granger and Wankel 2016). In addition, high DO in the water column ruled out the possibility of pelagic denitrification.

Careful examination of the Keeling plots for individual streams (Fig. 6) revealed that during the wet periods, $NO_3^-$ concentrations were significantly and linearly correlated with $1/[NO_3^-]$ in all the streams. These relationships strongly suggest mixing between two sources (with distinctive isotopic signatures) as the dominant process regulating the isotopic composition of the residual $NO_3^-$ in the streams during the wet periods. The different trends in the Keeling plots (Fig. 6) for individual streams indicate that the isotopic signature of the dominant $NO_3^-$ source varied temporally and spatially across the catchments. Negative trends on the Keeling plots for Bunyip, Lang Lang and Toomuc (Fig. 6) clearly show that the dominant $NO_3^-$ source was isotopically enriched (above +10‰ for Bunyip and Toomuc and +14‰ for Lang Lang) while the positive trends on the Keeling plots for Bass and Watsons show that the dominant $NO_3^-$ source was more isotopically depleted (less than +8‰ for Bass and less than +9‰ for Watsons). Nevertheless, the isotopic signatures of the dominant source; indicated by the y-intercepts of the Keeling plots were a lot more enriched than the initial $\delta^{15}N$-$NO_3^-$ of all three pre-identified $NO_3^-$ end members. Interestingly, these $\delta^{15}N$-$NO_3^-$ values increased with percentage agriculture except for Bass (Fig. 7). The fact that there was a clear fractionation pattern (~2:1) when integrating the isotope values of all the streams (catchment scale) suggests that denitrification was still prevalent during the wet periods (Fig. 8a) but this process was more likely to occur prior to $NO_3^-$ entering the streams via surface runoff. We explain these observations on the basis that increased rainfall created a 'hot moment' in the soil whereby organic matter mineralisation and nitrification were stimulated followed by denitrification within the waterlogged soil. Waterlogging can result in root anoxia and increased denitrification; leading to significant isotopic enrichment of the residual $NO_3^-$ (Chien et al. 1977, Billy et al. 2010) which was then washed into the streams. The extent of

this process (mineralisation – nitrification – denitrification) was greatest at Bass and Watsons; sites with the highest agricultural activity (Fig. 8a). Based on Fig. 8a, the isotope enrichments of the riverine $NO_3^-$ followed the denitrification trend of the artificial fertiliser and the $NO_3^-$ isotopes were distributed in between the denitrification ranges of both artificial fertiliser and SOM suggesting the important contribution of these two sources during the wet periods.

5          In fact, the deviation of the $\delta^{18}O$-$NO_3^-$:$\delta^{15}N$-$NO_3^-$ from the 1:1 trend to 2:1 corroborates the co-existence of other processes in our system (i.e. nitrification and/or anammox) in addition to denitrification. Based on the multi process model developed by Granger and Wankel (2016), the negative deflation of the denitrification trend (1:1) is strongly driven by concurrent $NO_3^-$ production catalysed by nitrification and/or anammox (Granger and Wankel 2016) when the rate of $NO_3^-$ reduction to $NO_2^-$ (via denitrification) is higher than the rate of $NO_2^-$ oxidation to $NO_3^-$ (via nitrification and/or anammox).

Higher reduction rate of $NO_3^-$ to $NO_2^-$ tends to create a $NO_2^-$ pool with enriched $\delta^{15}N$ due to isotopic fractionation (0‰ to 20‰) during the reduction of $NO_2^-$ to $N_2$ (the last step of denitrification). The subsequent oxidation of the $\delta^{15}N$-enriched $NO_2^-$ leads to the production of $NO_3^-$ which is isotopically more enriched than denitrified $NO_3^-$ owing to inverse kinetic fractionation effects (-35‰ to 0‰); driving the negative deviation of $\delta^{18}O$-$NO_3^-$:$\delta^{15}N$-$NO_3^-$ from the 1:1 trend (Granger and Wankel 2016). During the wet periods, simultaneous occurrence of these three processes (nitrification, annamox and denitrification) was

plausible due to the redox dynamics in the waterlogged soil zone. Downward percolation of oxygenated rain water could induce nitrification while denitrification and anammox could be promoted in the anoxic interstitial spaces of the waterlogged soil zone.

**4.4 Key controlling processes of nitrate during the dry periods**

Unlike the wet periods, only $NO_3^-$ in the Bass River showed an apparent relationship with $\delta^{15}N$-$NO_3^-$ (Fig. 6) during the dry

periods. There was no obvious relationships between $\delta^{15}N$-$NO_3^-$ and 1/[$NO_3^-$] for all other systems during the dry periods limiting the interpretation available from the Keeling plots. This also suggests that mixing between two end members alone is inadequate to explain the variability of $\delta^{15}N$-$NO_3^-$ during the dry periods. In general, during the dry periods, none of the samples show a noticeable pattern of denitrification on a biplot of $\delta^{18}O$ vs. $\delta^{15}N$ (Fig. 8b). The isotopic composition of the riverine $NO_3^-$ appeared to be clustered into three groups (A, B and C in Fig 8b):

(1)    $NO_3^-$ in group A showed consistent $\delta^{18}O$ but variable $\delta^{15}N$. This is demonstrated by the Lang Lang and Bass; coincident with the highest percentage of agriculture. The consistent $\delta^{18}O$ ($\delta^{18}O$ of ~2.5‰) shows the importance of nitrification ($\delta^{18}O$ of ~ -2.03 to -0.23‰) and at the same time ruled out the occurrence of denitrification and assimilation. In the absence of the removal processes, the heavy and variable $\delta^{15}N$-$NO_3^-$ values (+6‰ to +20‰) imply that animal manure was an apparent source of $NO_3^-$ during the dry periods for Lang Lang and Bass. This is

30          because volatilization of $^{14}N$ ammonia from the animal manure over time can lead to enrichment of $^{15}N$ in the residual $NH_4^+$ to > +20‰ (Batman and Kelly 2007) which can subsequently be nitrified to produce isotopically-enriched $NO_3^-$ without affecting its $\delta^{18}O$-$NO_3^-$. Tight coupling between mineralisation and nitrification results in $NO_3^-$ retaining the isotopic signature of the residual $NH_4^+$ (Deutsch et al. 2009) in the manure. Hence, it is not

surprising that $\delta^{15}N\text{-}NO_3^- > +13‰$ in the group A dataset is indicative of nitrified 'aged' animal manure. Because of the huge variability in the fractionation effect of ammonia volatilisation, it is difficult to affix an average $\delta^{15}N$ value to represent the isotopic signature of this end member. As such, apportioning the relative contribution of nitrified manure versus other sources (nitrified organic matter in the sediment and inorganic fertiliser) is not possible.

(2) $NO_3^-$ in group B has variable $\delta^{15}N$ and $\delta^{18}O$ values as shown by Bunyip and Toomuc. This could be attributed to isotopic fractionation during plant and/or algae uptake of $NO_3^-$ as substantiated by the parallel increase of $\delta^{18}O\text{-}NO_3^-$ versus $\delta^{15}N\text{-}NO_3^-$ (Fig. 9). Denitrification was ruled out due to high levels of dissolved oxygen in the water column. Close convergence of the linear relationships onto the theoretical assimilation trends of the nitrified artificial fertiliser and SOM (Fig. 9) reiterate the dominant contribution of these sources to the riverine $NO_3^-$ during the dry periods. It is worth noting that the initial $\delta^{18}O$ of nitrified $NO_3^-$ was estimated assuming full O isotopic equilibration between $NO_2^-$ and $H_2O$. Partial O isotope disequilibrium which was possible could affect the initial $\delta^{18}O$ signature of nitrified $NO_3^-$. If this happened, the minimum estimate of $\delta^{18}O$ of nitrified $NO_3^-$ could be more depleted and the overall linear relationship of $\delta^{18}O\text{-}NO_3^-:\delta^{15}N\text{-}NO_3^-$ would shift, resulting in more obvious contribution of artificial fertiliser, SOM and possibly organic fertiliser (Fig, 9). This scenario emphasizes the sensitivity of the initial $\delta^{18}O$ of nitrified $NO_3^-$ in determining the relative contribution of multiple sources in the catchment.

(3) $NO_3^-$ in group C comprised the most enriched $\delta^{15}N$ and $\delta^{18}O$ in the entire dataset (Fig. 8). These isotope values were observed in Watsons Creek which has the highest percentage of market gardens. These samples were collected when the creek was not flowing, hence the enriched $\delta^{15}N$ and $\delta^{18}O$ values could be indications of repeated cycles of internal processes (i.e. volatilisation, nitrification, denitrification and assimilation) in the same pool which enriched the N isotope but had slight effects on the O isotope of $NO_3^-$.

Although the isotope values during the dry periods appeared to be more likely controlled by artificial fertiliser and SOM, the contribution from organic fertiliser cannot be excluded. As mentioned in the preceding text, most of the fertiliser-derived $NO_3^-$ was denitrified in the catchment during the wet periods creating an artefact of heavy $NO_3^-$ isotopes in the streams. This $NO_3^-$ could exhibit a similar enriched isotopic composition as the volatilised manure (particularly if there was disequilibrium of O isotope between $NO_2^-$ and $H_2O$). Overlapping of these isotopic values made it difficult to distinguish between all the three sources – a disadvantage of using $NO_3^-$ isotopes in a system where multiple sources and transformation processes coexist.

## 5 Conclusions

This study highlights the effect of rainfall conditions on the predominance of sources and transformation processes of $NO_3^-$ on both individual stream and catchment scale. The significant positive relationships between percentage agriculture and $NO_3^-$ concentrations ($r^2 = 0.29$; $p<0.05$) as well as $\delta^{15}N\text{-}NO_3^-$ ($r^2 = 0.39$; $p<0.01$) particularly during the wet period showed that enriched $NO_3^-$ concentrations and $\delta^{15}N\text{-}NO_3^-$ values resulted from agricultural activities. The dual isotopic compositions of

$NO_3^-$ revealed that both mixing of diffuse sources and biogeochemical attenuation controlled the fate of $NO_3^-$ in the streams of the Western Port catchments. During the wet periods, inorganic fertiliser appeared to be the primary source of $NO_3^-$ to the streams while SOM, in addition to inorganic fertiliser was also a dominant source of $NO_3^-$ during the dry periods. Denitrification in the catchment appeared to be the more active removal process during the wet periods in contrast to a greater

importance of in-stream assimilation during the dry periods. However, these removal processes were insufficient to remove the agricultural-derived $NO_3^-$ inferring that the streams were unreactive conduits of $NO_3^-$ which might pose a potential $NO_3^-$ enrichment threat to downstream ecosystems. To the best of our knowledge, this is the first study in Australia and also one of the very few targeted studies in the southern hemisphere investigating the origin and sink of $NO_3^-$ on a catchment scale using both $\delta^{15}N$ and $\delta^{18}O$ of $NO_3^-$. The application of $NO_3^-$ isotopes in a region with highly variable and unpredictable rainfall patterns

such as the Western Port catchments although challenging; is imperative particularly in setting guidelines for sustainable land use management actions.

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

**Table 1:** The isotopic compositions of potential sources of $NO_3^-$ in the catchment

| Sample | $\delta^{15}N$-TN (‰) | $\delta^{18}O$-H$_2$O (‰) |
|---|---|---|
| Artificial/inorganic fertiliser | -0.5 to +0.7 | - |
| Cow manure/organic fertiliser | +6 to +13 | - |
| Sediment (SOM) | +4 to +5 | - |
| Stream water | - | -5.5 to -4.9 |

5  **Table 2:** Comparison of $NO_3^-$ concentrations and isotopes across different systems reported in the literature

| Study area | Percentage agriculture (%) | $[NO_3^-]$ (µM) | $\delta^{15}N\text{-}NO_3^-$ (‰) | $\delta^{18}O\text{-}NO_3^-$ (‰) | Reference |
|---|---|---|---|---|---|
| Mississippi River Basin, USA | 0 to 100 | 3.6 to 1290 | -1.4 to +12.3 | +3.1 to +43.3 | Chang et al. 2002 |
| Connecticut River Watershed, USA | 0.8 to 52 | 0 to 360 | *0 to +14.5 | *-2 to +14 | Barnes et al. 2010 |
| New York, USA | 0 to 72 | *5 to 640 | *0 to +9 | *-8 to +40 | Burns et al. 2009 |
| Mid-Atlantic and New England states of the USA | 2 to 38 | 7.9 to 184 | +3.6 to +8.4 | +11.7 to +18.5 | Mayer et al. 2002 |
| Baltic Sea catchment | 1 to 81 | 3 to 216 | -1.5 to +14 | +10 to +25 | Voss et al 2006 |
| Trout River catchment, Atlantic Canada | ~39.7 | 32 to 170 | +2.13 to +6.35 | +1.51 to +7.07 | Danielescu and MacQuarrie 2013 |
| Skuterud catchment, Norway | 0 to 100 | 21 to 1850 | +3 to +18 | +10 to +24 | Kaste et al. 2006 |
| Mørdre catchment, Norway | 74 to 100 | 120 to 2320 | +8 to +15 | +5 to +20 | Kaste et al. 2006 |
| Pearl river drainage basin | ~86 | 41 to 110 | +1.9 to +17.6 | +5.6 to +17.3 | Chen et al. 2009 |
| Westernport catchment, Australia | 2 to 96 | 4 to 790 | +5.7 to +33 | +1.4 to +12.7 | This study |

*Values estimated from presented figures, might not accurately represent the actual data

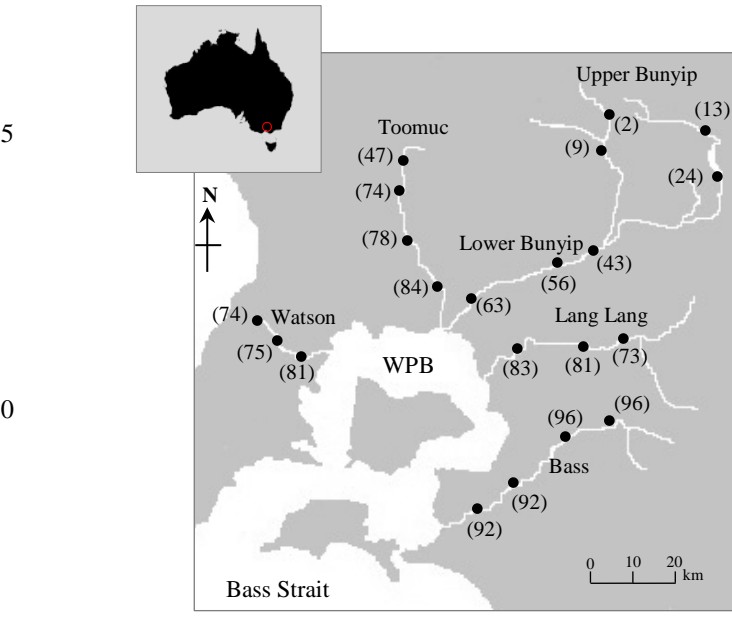

**Figure 1: Map of Western Port Bay (WPB) in southern Victoria, Australia and major rivers discharging into WPB. Closed circles represent sampling sites where surface water samples were obtained. Values in parentheses represent the % agriculture area in the catchment.**

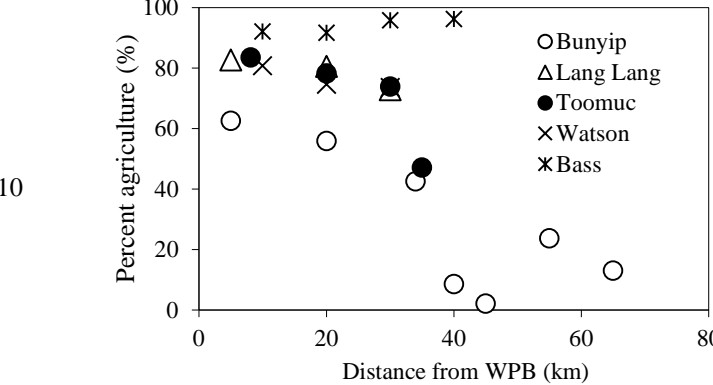

Figure 2: The percent agriculture for each of the sampling sites.

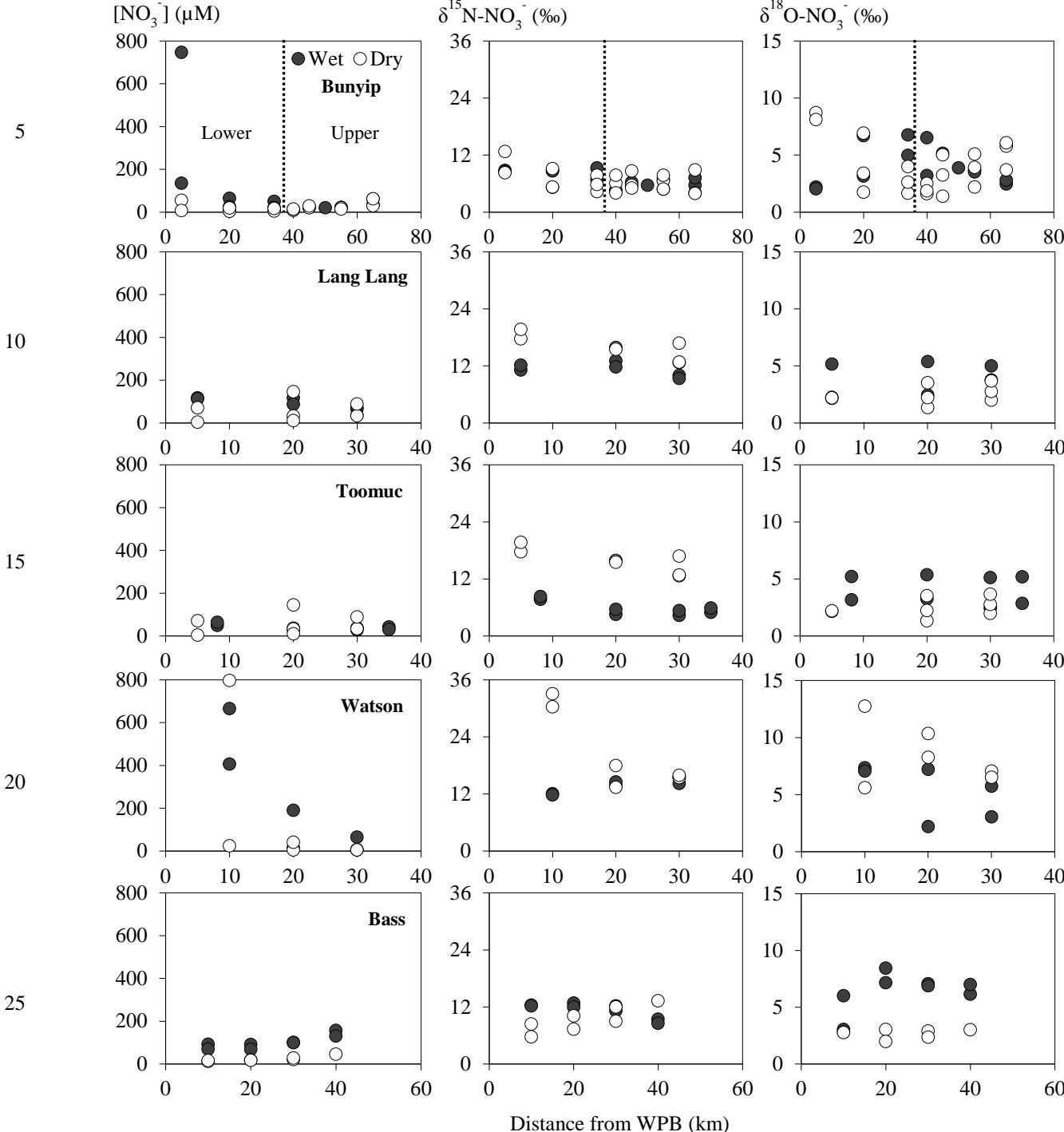

**Figure 3: Spatial and temporal variations of nitrate concentrations and isotopes values. Closed circles represent data obtained during the wet periods. Open circles represent data obtained during the dry periods.**

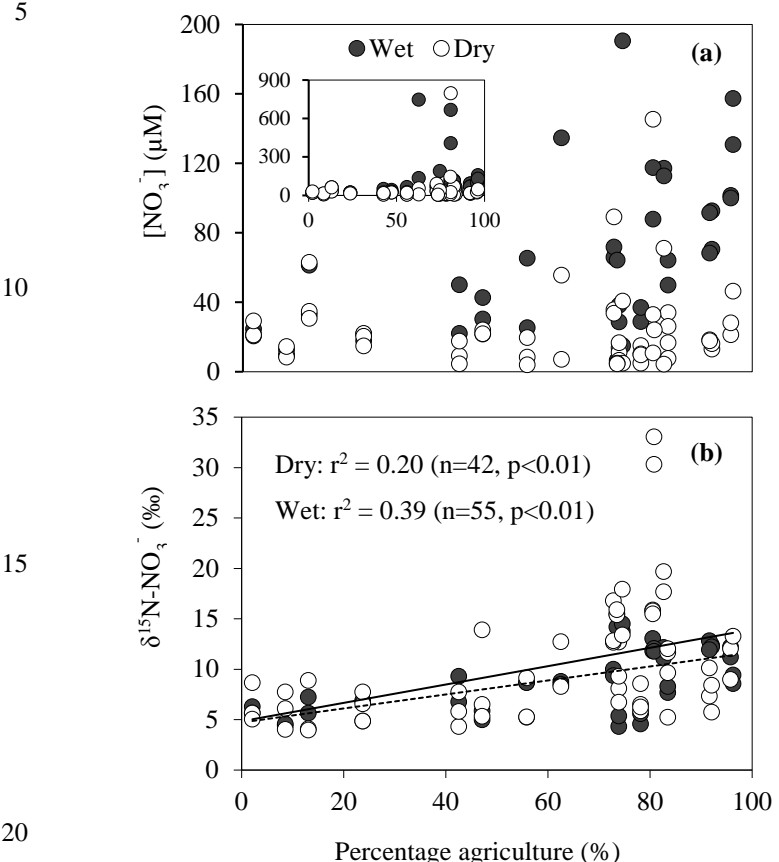

Figure 4: Relationship between (a) NO₃⁻ concentration; (b) δ¹⁵N-NO₃⁻ and the percentage of agricultural land use. In (b) solid line represents the relationship between the variables during dry periods; dotted line represents wet periods.

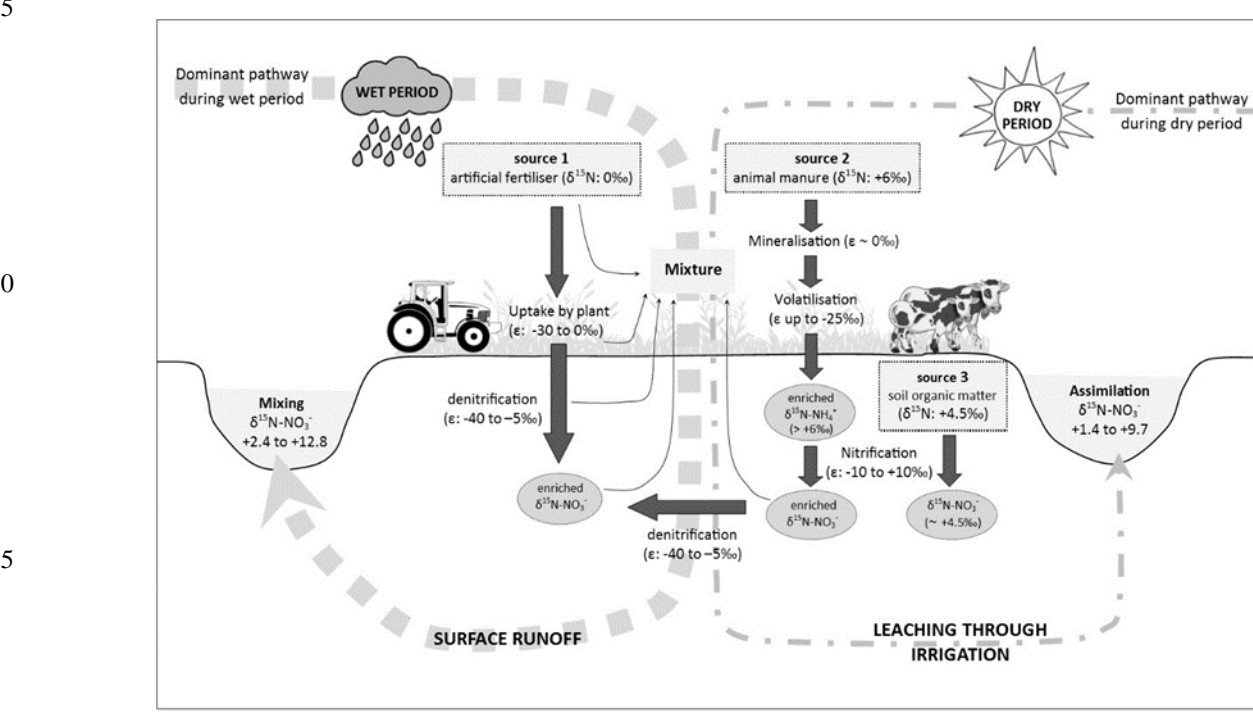

**Figure 5: Conceptual diagram illustrating the sources and processes of NO₃⁻ during the wet and dry periods in the Western Port catchment. The values of enrichment factor (ε) were obtained from the literature (Kendall et al. 2007) to indicate the relative contribution of the transformation processes to the isotopic compositions of the residual NO₃⁻.**

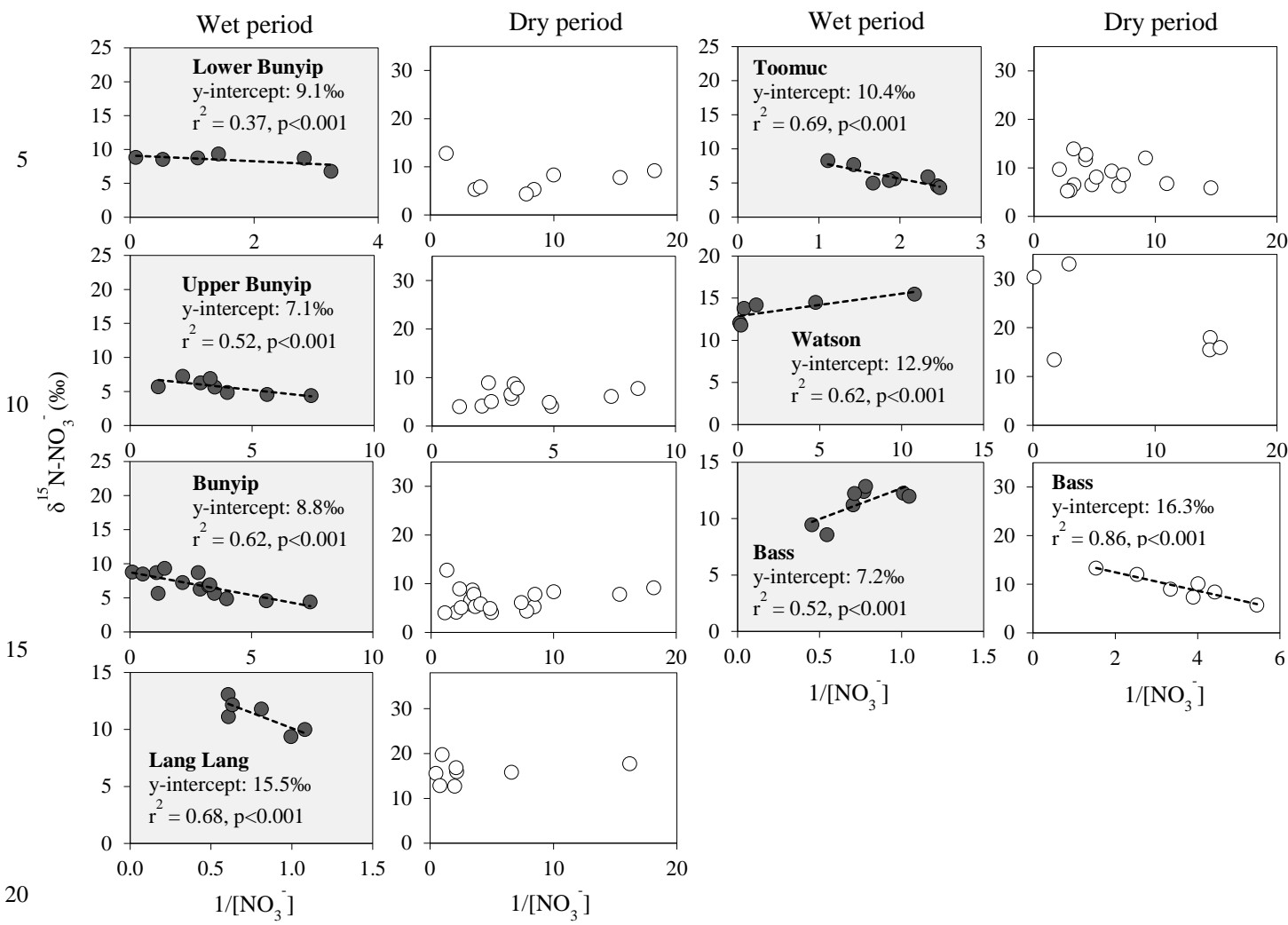

**Figure 6: Relationship between δ¹⁵N-NO₃⁻ and 1/[NO₃⁻] for individual streams during the wet and dry periods.**

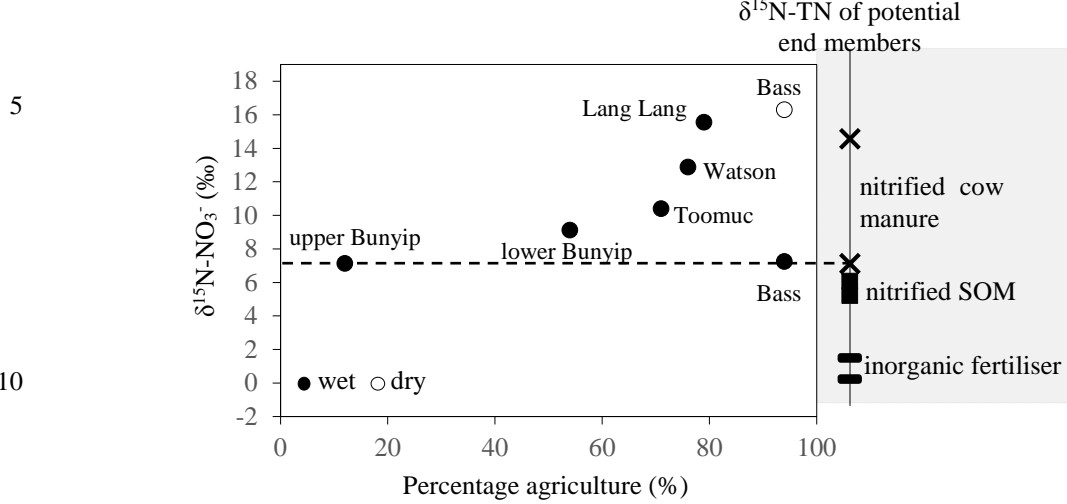

**Figure 7: Relationship between δ<sup>15</sup>N-NO₃⁻ of the dominant initial source (indicated by the y-intercept of the Keeling plots; Figure 6)**

 **and percentage agriculture during wet periods. Data for Bass-dry period was also presented because only the Keeling plot for Bass-dry period indicates mixing between different sources. The shaded area represents the δ<sup>15</sup>N-TN of the potential end members.**

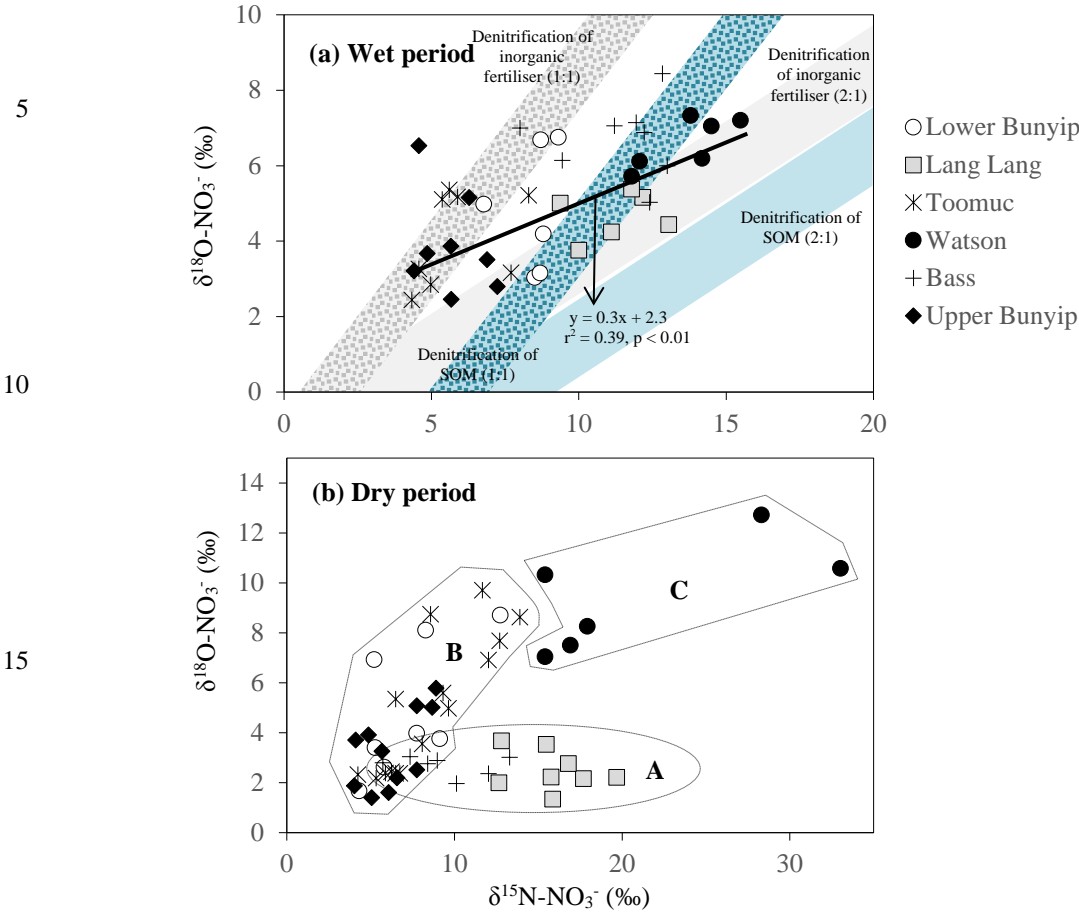

**Figure 8: Biplot of $\delta^{15}N\text{-}NO_3^-$ versus $\delta^{18}O\text{-}NO_3^-$ for (a) wet and (b) dry periods. Blue shaded area represents possible isotopic compositions of denitrified $NO_3^-$ originated from SOM ($\delta^{15}N$: +4.5‰). Grey shaded area represents the possible isotopic composition of denitrified $NO_3^-$ originated from inorganic fertiliser ($\delta^{15}N\text{-}NO_3^-$: +0.1‰). The $\delta^{18}O\text{-}NO_3^-$ used were -2.3‰ and +-0.23‰ representing the minimum and maximum estimates of $\delta^{18}O$ of nitrified $NO_3^-$, respectively. The shaded area were plotted based on the theoretical 1:1 and 2:1 denitrification relationships between $\delta^{15}N\text{-}NO_3^-$ and $\delta^{18}O\text{-}NO_3^-$ (Kendall et al. 2007).**

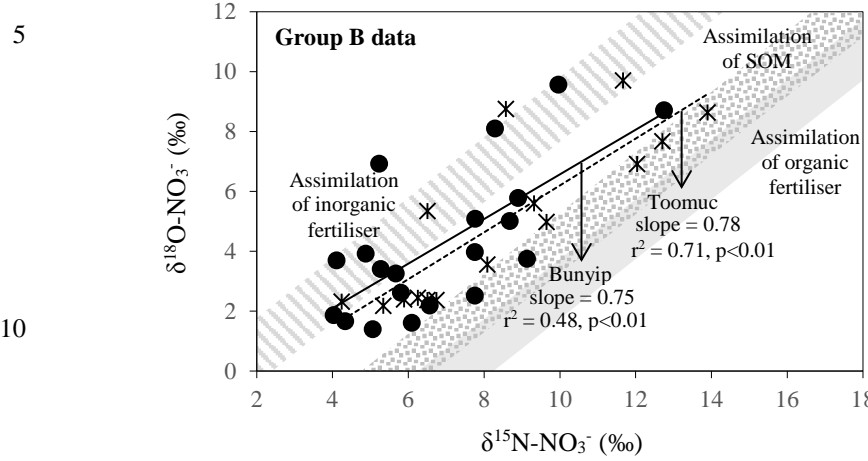

**Figure 9: Biplot of δ¹⁵N-NO₃⁻ versus δ¹⁸O-NO₃⁻ for Bunyip and Toomuc (group B data in Fig. 8b). Shaded areas represent theoretical assimilation trends for cow manure, SOM and inorganic fertiliser. The maximum and minimum starting values for δ¹⁸O-NO₃- were estimated from Equation 1. The starting δ¹⁵N-NO₃⁻ is the δ¹⁵N-TN value of respective end member. Solid and dotted lines represent the assimilation trends for Bunyip (both lower and upper Bunyip) and Toomuc, respectively. Assimilation rather than denitrification was considered a more plausible process controlling the distribution pattern for the group B dataset as the water column was oxic throughout the study period.**