# Peer review of "Stable isotopes of nitrate reveal different nitrogen processing mechanisms in streams across a land use gradient during wet and dry periods"

_Biogeosciences, 2017_

## Referee Comment (RC1) · Anonymous Referee #2 · 2 Nov 2017

Wong and co-authors present a study of stable isotopes of nitrate from five streams within the same catchment area in Southeast Australia sampled during wet and dry periods. The five streams show different degrees of land use intensities. The aim is to reveal different sources and transformation processes of nitrate compared between rainfall patterns through the isotopic composition ($d15N$-$NO_3$- and $d18O$-$NO_3$-). Results show that differences between wet and dry periods can be explained by the dominance of different sources on the isotopic composition. During wet periods artificial fertilizer was probably the main source, whereas nitrified organic matter in sediment and nitrified manure dominated the sources during dry periods. The manuscript is well written and presents the results in a logical order. The figures illustrate the findings

very well. This novel dataset is suitable for publication in Biogeosciences, however, there are some points that should be addressed by the authors.

Main points:

1) To study the impact of rainfall on isotopic composition in a more rigorous way, it would have been interesting to use samplings with different amounts of rainfall in the previous days (instead of only differentiating between wet and dry periods) in order to see whether rainfall and isotopic composition could be correlated. At least the authors should explain why such a study was not carried out.

2) Could the data not have been explored more thoroughly, e.g. other statistical methods than linear regression in order to identify multiple sources? What about isotope mixing and emission modeling for source identification?

3) Significant correlations with very low r2 for isotopic composition and % agriculture (Fig. 4) are used as argument for "dominance of anthropogenic nitrogen inputs within the catchment". The discussion should include a more detailed comparison to studies which found a similar but much stronger correlation between d15N of nitrate and land use.

4) Is there any information to take away from individual samplings within the same stream? There is no information on river flow rates, for example. Could patterns of isotope data within the streams be explained by mixing of sources or in-stream processing?

Minor points:

Page 2 line 11 : Kendall 2007 ; "et al." is missing

Page 2 line 21-24: please explain in more detail in which way rainfall patterns are different in the southern hemisphere compared to the northern hemisphere.

Page 2 line 28: please delete space in "samplin g".

**BGD**

Page 3 line 21-23: what are the criteria to give the amount of rainfall for 5 days (5-10 days) before sampling of wet periods (dry periods)? Following up, what is the residence time of water in the aquifer and in the river?

Page 4 line 10: a figure with some additional water quality parameters would be nice to include as a supplementary material.

Page 4 line 16-18: how many samples of fertilizer and cow manure were analysed? Please specify.

Page 6 line 14/15: what about atmospheric deposition? It is only mentioned on page 7 line 1. Couldn't mixing lead to a depletion of the d18O, with NO3- from atmospheric deposition still contributing partly to the signal?

Page 6 line 27: delete the d of "comprised".

Page 7 line 8-10: there is no statistic evidence given by the authors to show that there is an actual trend, so this should be rephrased.

Page 7 line 10-12: for the dry periods there are at least 6 data points with NO3- conc. > 50 $\mu$M, so "consistently lower" (than 36 $\mu$M) is not correct.

Page 7 line 13: replace "entered" by "entering".

Page 7 line 21/22: please state clearly, that although significant, correlation coefficients r2 are 0.2 and 0.39, respectively, so quite low. From there on it is obvious that the relationship between d15N and % agriculture is not evident at all from this study. This has to be expressed more clearly.

Page 7 line 24-26: the comparison to the other studies has to be made more in detail. For example, Voss et al (2006) observed a significant correlation for 11 streams and weighted monthly means of d15N and % agriculture. If comparable to this data, it should be Figure 7 from this study (if it is correct that it represents average values per stream). And for this representation, there is no significant correlation.

Page 8 line 18-20 (and following sentences): Give first all the evidence that allows you to conclude that in-stream processing was not the dominant process for regulating the isotopic composition. These arguments could be supported by a more detailed discussion of the relevant literature.

Page 8 line 20-22: Put figures of d18O vs [NO3-] and d18O vs d15N for individual streams in supplementary material to support your argument.

Page 9 line 23: add "be" in between "subsequently" and "nitrified".

Page 10 line 19-21: as stated above, according to figure 4 there is no significant correlation for [NO3-] and percentage agriculture and r2 for the corr. between d15N and percentage agriculture are low, so please rephrase this conclusion. Similarly please rephrase the related sentence in the abstract (page 1 line 17/18).

Figures

Fig. 1: indicate percentage agriculture for each sampling site.

Fig. 2: For the Watsons river the "distance from WPB" does not correspond to the values from Fig. 3 (max= 30 km).

Fig. 4: use A and B for the two panels. In the lower panel indicate which trend curve corresponds to which dataset.

Fig. 7: Are these average values per site? If so, please indicate the SD. For Bass river (dry period) the value is somewhat high compared to Fig. 6. Please explain.

---

## Author Comment (AC1) · 13 Nov 2017

We thank the reviewer for the constructive comments.

Major comments:

(1) There was no correlation between the isotope values (both $\delta15N$ and $\delta18O$) and the total amount of rainfall for 10 days before each sampling event as shown in Fig. 1. This kind of relationship might be expected for pristine environment, however for heavy anthropogenic-affected environment like the WP catchment; it is impossible to observe such correlation because of the dominance of other sources of nitrate. In this study,

the condition of the soil (i.e. wet versus dry) in the catchment and the residence time of the rivers were affecting the occurrence and the extent of certain biological processes in the catchment and thus the isotope values of the residual nitrate rather than rainfall. We believe this has been shown nicely in the manuscript. The amount of rainfall in this study was used as a direct indicator of soil condition and residence time of the rivers, hence why the isotope data was grouped into wet and dry periods instead of the actual amount of rainfall. Fig. 1, however; is a strong evidence to show that the amount of rainfall had minimal effect on the isotope values in this study; supporting our contention on the insignificant direct contribution of rainfall amount to the overall nitrate dynamic in the catchment. Fig. 1 will be included in the supplementary information in the revised manuscript.

(2) Apportioning the contribution of multiple sources of nitrate using the suggested models requires a well-defined isotope values of the end members, accurate fractionation factors of the possible processes (i.e. denitrification and mineralisation) as well as the loads and rates of each possible source and process particularly for an emission model. Unfortunately, determining the fractionation factors of the processes was beyond the scope of this study. These models were not suitable in this study because there was significant overlapping of the end member nitrate isotope values as well as the lack of information on the rates of different types of fertiliser applications. Hence, only a qualitative assessment of the sources was presented in this study.

(3) We agree with the reviewer and will include more detailed comparisons to studies with similar findings in the revised manuscript.

(4) Individual samplings within the same stream did not provide sufficient data points for the interpretation on the processes governing the isotope values of the residual nitrate in the streams. For example, at Watson creek only three samples were obtained during each sampling trip. We are not confident to deduce any findings based on that even though the isotope biplot or the keeling plots for some of the sampling events showed significant correlations.

All the minor comments will be addressed in the revised manuscript.

[Figure]

Figure 1: Relationship between (a) $\delta^{15}N-NO_3$-; (b) $\delta^{18}O-NO_3^-$ of the streams and the total
amount of rainfall for 10 days before the sampling event.

---

## Referee Comment (RC2) · Anonymous Referee #3 · 15 Feb 2018

In the manuscript Stable isotopes of nitrate reveal different nitrogen processing mechanisms in streams across a land use gradient during wet and dry periods, Wong et al present natural abundance nitrate isotopes from five streams across a land use gradient during wet and dry seasons, allowing them to elucidate the controls on sources and transformations of nitrate. This is an interesting dataset and the authors have been resourceful and knowledgeable in their presentation and interpretation of the data. However, prior to publication the manuscript would benefit from a clear and concise definition of terms, and a clearer explanation of the isotope effects and there subsequent implications, as currently it seems hard to follow in places for the none expert reader.

An important aspect of the interpretation of this dataset is that there is a tight coupling of mineralisation and nitrification, resulting in no isotope effect being expressed and hence the 15N of organic matter / ammonium and nitrate are similar. Currently this is not fully explained until Page 9 Line 24, making it difficult to understand the authors interpretation of the data prior to this, explaining this earlier on in the discussion will enable the reader to follow your thoughts / interpretation. A good example of this is Page 6 Line 15/16, break this thought down and explain to the reader here the tight coupling between mineralization and nitrification and hence no isotope effect being observed.

Specific comments

Page 2 Line 10: it would be valuable here to state that you are talking about kinetic isotope effects and not equilibrium.

Page 2 Line 23 to 25: for the none expert, please explain why rainfall patterns are different in the southern hemisphere and its subsequent effects.

Page 2 Line 32 to 33: start preparing your reader here, why are denitrification and assimilation more prevalent in wet periods.

Study area: throughout this section you refer to the gradient in land use across the catchment, a map of this would be a great addition to the manuscript (or could maybe be added to Figure 1).

Page 3 Line 32: how do the authors think using an integrated signal could of biased their interpretation of the results?

Page 5 Lines 9 to 10: the term total nitrogen needs to be defined here, as it is important for the mineralization discussion later on. Would particulate organic nitrogen not be a more suitable term? Also, please add in that the values are relative to AIR and the precision of the measurements.

Page 5 Lines 24: Please add in nitrite concentrations, to confirm for the reader that the

values are less than 1% of the nitrate (as stated in the methods).

Page 6 Line 3: Be clear that you are talking about 15N values here.

Page 6 Line 4: The enriched 15N-nitrate values seem to be constrained to a thin band between 70 to 85% agriculture, but then drop away again at higher percentage agriculture, do the authors have any hypotheses for this?

Page 6 Line 5: Surely the same is true for the Bass.

Page 6 Line 16 to 28 and Equation 1: It would be valuable to explain to the readers the value of using both N and O isotopes i.e. N is recycled between fixed N pools and the O atoms are removed and then replaced by nitrification and thereby sensitive to internal processing (this could come here or in the introduction). The authors need to discuss the more recent literature when introducing and determining the oxygen isotope signal imparted by nitrification, the work of Carly Buchwald is particularly pertinent here.

Page 6 Line 29: Is this value for cow manure similar to the literature to date?

Page 7 Line 14: 'terrestrial' what are the authors referring to here, fertilizer, manure, leaf litter, please be consistent with the use of terms throughout.

Page 7 Lines 15 onwards: a slight restructure here would be beneficial, you are presenting your conclusions before the evidence, discussing your isotope data first in this section would make it easier to follow.

Page 7 Line 21: I think the authors are referring to Table 2 here.

Page 8 Line 15 and 32: I strongly suggest the authors cite and discuss the implications of the outcomes from the work of Granger and Wankel, 2016 (Isotopic overprinting of nitrification on denitrification as a ubiquitous and unifying feature of environmental nitrogen cycling; PNAS) and how this may influence your interpretation of N turnover in your catchment.

Page 10 section (3): an earlier introduction of the different behaviors of N and O isotopes during internal processing of nitrate will make this section easier to understand for the none expert reader. I would not put denitrification under the heading recycling, if the authors are referring to nitrate reduction, followed by reoxidation please say so.

Page 10 Line 11: Do the authors know when fertilizer is applied in this catchment, how does this align with your runoff / turnover hypotheses?

Figure 1: Please mark on the map of Australia where southern Victoria is.

Figure 3: Mark on upper / lower Bunyip.

Figure 5: Where have the authors taken these isotope effects from? Please cite the relevant literature in the caption. A positive / inverse isotope effect for nitrification?

Figure 7: More details are needed in the figure caption, what do the crosses and dashed line mean? I also assume that it is the y intercept values determined in Figure 6 that have been plotted.

Figure 9: define what starting values you have used and where they have come from, particularly for the oxygen isotopes.

―――――――――――――――――――――――――

---

## Author Comment (AC2) · 16 Mar 2018

We thank the reviewer for the constructive comments.

Reviewer #3 GENERAL COMMENTS In the manuscript Stable isotopes of nitrate reveal different nitrogen processing mechanisms in streams across a land use gradient during wet and dry periods, Wong et al present natural abundance nitrate isotopes from five streams across a land use gradient during wet and dry seasons, allowing them to elucidate the controls on sources and transformations of nitrate. This is an interesting dataset and the authors have been resourceful and knowledgeable in their presentation and interpretation of the data. However, prior to publication the

manuscript would benefit from a clear and concise definition of terms, and a clearer explanation of the isotope effects and there subsequent implications, as currently it seems hard to follow in places for the none expert reader. An important aspect of the interpretation of this dataset is that there is a tight coupling of mineralisation and nitrification, resulting in no isotope effect being expressed and hence the 15N of organic matter / ammonium and nitrate are similar. Currently this is not fully explained until Page 9 Line 24, making it difficult to understand the authors interpretation of the data prior to this, explaining this earlier on in the discussion will enable the reader to follow your thoughts / interpretation. A good example of this is Page 6 Line 15/16, break this thought down and explain to the reader here the tight coupling between mineralization and nitrification and hence no isotope effect being observed. We will add a few sentences to explain the minimal isotope effect of the combined reaction of mineralisation and nitrification as suggested by the reviewer. Page 6 Line 31: Nitrogen isotope of the NO3- produced from these end members usually retains the signature of the $\delta$15N-TN as a result of tight coupling between mineralisation (production of ammonium from organic matter) and nitrification (oxidation of ammonium to NO3-) as well as the minimal isotopic fractionation of both processes. It is well documented in the literature that in soil environment, mineralisation causes a small isotopic fractionation ($\pm 1$‰ $Kendall et al. 2007$) $to the produced NH4+. In agricultural areas where NH4+$ $is rapidly consumed or assimilated by crops, nitrification rate is usually low and would also exert as mall isotopic fractionation to$.

SPECIFIC COMMENTS Page 2 Line 10: it would be valuable here to state that you are talking about kinetic isotope effects and not equilibrium. This will be made clearer in the revised manuscript

Page 2 Line 23 to 25: for the none expert, please explain why rainfall patterns are different in the southern hemisphere and its subsequent effects. This will be explained in the revised manuscript. Page 2 Line 30: The southern hemisphere tends to have more sporadic and variable rainfall patterns compared to the northern hemisphere and
Australia is an example of this. The variable rainfall patterns can modulate different efficiencies of denitrification in soils and thus different fractionation effects to the residual NO3- pool.

Page 2 Line 32 to 33: start preparing your reader here, why are denitrification and assimilation more prevalent in wet periods. Study area: throughout this section you refer to the gradient in land use across the catchment, a map of this would be a great addition to the manuscript (or could maybe be added to Figure 1). Following sentences will be added to the revised manuscript to explain why denitrification can be more prevalent in wet periods. A land use map will be added as supplementary material (Figure S2). Page 3 Line 7: In some studies (e.g. Riha et al. 2014; Kaushal et al. 2011), denitrification and assimilation by plants and algae have been reported to be more prominent during the dry seasons compared to the wet seasons but in other studies (e.g. Murdiyarso et al. 2010; Enanga et al. 2016) denitrification appeared to be more prevalent during the wet seasons as precipitation induces saturation of soils resulting in oxygen depletion and thereby low redox potentials that favour denitrification.

Page 3 Line 32: how do the authors think using an integrated signal could of biased their interpretation of the results? We think the integrated signal could potentially bias the interpretation of the results; however, the integrated signal was the best representation of the percentage agriculture area in the catchment.

Page 5 Lines 9 to 10: the term total nitrogen needs to be defined here, as it is important for the mineralization discussion later on. Would particulate organic nitrogen not be a more suitable term? Also, please add in that the values are relative to AIR and the precision of the measurements. Particulate organic matter is a more suitable term however this was not specifically measured in our study. We used $\delta15N$ of total nitrogen of the soil to directly represent the soil organic portion as most of the nitrogen in soils is generally bound in organic forms. This will be explained more thoroughly in the revised manuscript.

Page 5 Lines 24: Please add in nitrite concentrations, to confirm for the reader that the values are less than 1% of the nitrate (as stated in the methods). Nitrite concentrations will be added to the results section of the revised manuscript. Page 6 Line 6: Nitrite concentrations ranged between $0.1\mu$mol/L and $0.4\mu$mol/L.

Page 6 Line 3: Be clear that you are talking about 15N values here. This sentence will be corrected to reflect more clearly on the $\delta$15N values. Page 6 Line 18: Overall, $\delta$15N of the riverine NO3- spanned a wide range (+4 to +33‰.

Page 6 Line 4: The enriched 15N-nitrate values seem to be constrained to a thin band between 70 to 85% agriculture, but then drop away again at higher percentage agriculture, do the authors have any hypotheses for this? We hypothesised that the drop off of $\delta$15N-NO3- at > 85% agriculture was due to recent and possibly over-fertilisation of NH4+ fertiliser resulting in active nitrification. As a large amount of NH4+ was available, oxidation of NH4+ to NO3- became the rate-determining step resulted in large fractionation (-38‰ to -14‰ $Casciottietal.2003) and depleted \delta$15N-NO3- in the residual NO3- pool. Unfortunately we could not test this hypothesis as we did not have the information on the rates of fertiliser application and nitrification. This would be a good avenue for future research.

Page 6 Line 5: Surely the same is true for the Bass. We agree with the reviewer and Bass will be included as exhibiting the same effect as Bunyip in the revised manuscript. Page 6 Line 20: Among all sites, $\delta$15N-NO3- values in the Bunyip and Bass were relatively depleted (+4 to +12‰ for Bunyip and +10 to 12‰ for Bass), with the lower range found at upper Bunyip (+4 to +8‰.

Page 6 Line 16 to 28 and Equation 1: It would be valuable to explain to the readers the value of using both N and O isotopes i.e. N is recycled between fixed N pools and the O atoms are removed and then replaced by nitrification and thereby sensitive to internal processing (this could come here or in the introduction). The authors need to discuss the more recent literature when introducing and determining the oxygen isotope signal

imparted by nitrification, the work of Carly Buchwald is particularly pertinent here. We will add a few sentences in the introduction to discuss the value of using both N and O isotopes. We will also discuss the calculation used to estimate $\delta$18O-NO3- imparted by nitrification in more detail. All the texts and figures in the manuscript have been revised to reflect on these changes. Page 2 Line 7: To date, the most promising tool to investigate the sources and sinks of NO3- are the dual isotopic compositions of NO3- at natural abundance level (expressed as $\delta$15N-NO3- and $\delta$18O-NO3- in ‰. Preferential utilisation of lighter isotopes (14N and 16O) over heavier isotopes (15N and 18O) leads to distinctive isotopic signatures that differentiate the various NO3- sources/end members (e.g. inorganic and organic fertiliser, animal manure, atmospheric deposition) and the predictable kinetic fractionation effect when NO3- undergoes different biological processes (e.g. nitrogen fixation and denitrification). For instance, denitrification and phytoplankton assimilation fractionate N and O isotopes in a 1:1 pattern. Simultaneous measurement of $\delta$15N-NO3- and $\delta$18O-NO3- also provides complementary information on the cycling of NO3- in the environment. $\delta$18O-NO3- is a more effective proxy of internal cycling of NO3- (i.e. assimilation, mineralisation and nitrification) compared to $\delta$15N-NO3-. This is because during NO3- assimilation and mineralisation, N atoms are recycled between fixed N pools and the O atoms are removed and replaced by nitrification (Sigman et al. 2009; Buchwald et al. 2012). Page 7 Line 6: The $\delta$18O of NO3- generated by nitrification of these sources is decoupled from $\delta$15N-NO3- but relies on the oxygen isotope of water ($\delta$18O-H2O), oxygen isotope of dissolved oxygen ($\delta$18O-O2) as well as the kinetic and equilibrium isotope effects during the sequential oxidation of NH4+ to NO2- then NO3- (Casciotti et al. 2010; Buchwald et al. 2012). Previous culture studies (Casciotti et al. 2010; Buchwald and Casciotti 2010; Buchwald et al. 2012) and observations in various marine systems (Sigman et al. 2009; Granger et al. 2013; Rafter et al. 2013) have found that $\delta$18O values for nitrified NO3- were within a few ‰ of the $\delta$18O-H2O. Hence, -5.3‰ $the\ average\ value\ of\ \delta$18O-H2O is adopted to represent the lower estimate of $\delta$18O of the nitrified NO3- in this study. In a system where equilibrium exchange of oxygen between H2O and NO2- is negligible

but respiration and denitrification are prevalent/co-occurring, $\delta$18O-NO3- can be much greater than that of $\delta$18O-H2O. In this study, the $\delta$18O-NO3- values were all more enriched than -5.3‰ suggesting the co-occurrence of a fractionating process, most likely denitrification (this is discussed in the following section). Based on this reason, using -5.3‰ can potentially underestimate the $\delta$18O of the nitrified NO3-. The conventional 2:1 ($\delta$18O-H2O:$\delta$18O-O2) fractional source contribution model (Equation 1) is therefore used to calculate the maximum estimate of $\delta$18O of the nitrified NO3- in our study which is +4.3‰ by using -5.3‰ for the average $\delta$18O-H2O and +23.5‰ for $\delta$18O-O2.

Page 6 Line 29: Is this value for cow manure similar to the literature to date? Yes these values are similar to the literature to date.

Page 7 Line 14: 'terrestrial' what are the authors referring to here, fertilizer, manure, leaf litter, please be consistent with the use of terms throughout. This term refers to a combination of sources and this will be made clearer in the revised manuscript. Page 8 Line 22: …..in stream NO3- comprised mainly of terrestrially derived NO3- (i.e. inorganic fertiliser, manure and soil organic matter) entered the streams through surface runoff…

Page 7 Lines 15 onwards: a slight restructure here would be beneficial, you are presenting your conclusions before the evidence, discussing your isotope data first in this section would make it easier to follow. This section will be restructured as suggested by the reviewer. Page 8 Line 2: Agricultural land use (i.e. market gardens and cattle rearing) appeared to influence NO3- concentrations in our study sites. As shown in Fig. 4(a), during the wet periods, high NO3- concentrations (> 40 $\mu$M) were particularly observed at sites with more than 70% agricultural land use. During the dry periods, although NO3- concentrations were generally lower than 36$\mu$M, the outliers were observed at sites with more than 70% agricultural land use. Similarly, enriched $\delta$15N-NO3- in the streams were mainly found at sites with high percentage agricultural land use (between 75 to 85%) for both dry and wet periods suggesting that enriched $\delta$15N-NO3- in the stream were originated from agricultural activities. In fact,

the most enriched $\delta$15N-NO3- values (>30‰ were observed at the most downstream site of Watson Creek which has the largest percentage of market gardens (although the total agricultural area is not the highest amongst all the studied sites). We also observed a significant positive relationship between $\delta$15N-NO3- and percentage agriculture during the wet periods (Fig. 4b) which further supports the contention that agricultural activities were the main control of the $\delta$15N-NO3- in the streams. Other researchers have also documented similar trends of enriched $\delta$15N-NO3- with increasing percentage agriculture. For example Harrington et al. 1998, Mayer et al. 2002 and Voss et al. 2006 observed highly significant positive relationships between percentage agriculture land area and $\delta$15N-NO3- with r2 $\sim$ 0.7. However, these studies showed comparatively narrower and more depleted ranges of $\delta$15N-NO3- with 2.0 to 7.3‰ $4 to 8\ and\ -0.1 to 8.3\ respectively, suggesting more subtle changes in \delta$15N-NO3- over a large span of agriculture land areas in these studies compared to our study. Given that none of the predicted sources of NO3- in the Western Port catchment exhibited an initial $\delta$15N-NO3- of more than +6‰ the isotopically-enriched NO3- as well as the variability of NO3- concentrations observed in this study were consequences of a series of transformation processes. Hence, we propose the following factors to explain the heavy isotopes and the different NO3- concentrations across different periods observed in our study: (1) During the wet period when surface runoff was conspicuous and residence time of the water column was low, in-stream NO3- comprised mainly of terrestrially derived NO3- (i.e. fertilisers, manure and soil organic matter) and there was limited in-stream processing of these NO3-. The high NO3- concentrations and the heavy $\delta$15N-NO3- values reflect the occurrence of mineralisation, nitrification and subsequent preferential denitrification of the isotopically lighter NO3- source/s in either the waterlogged soil or in the soil zone underneath the market gardens before transport to the streams through surface runoff. (2) During the dry periods when surface runoff was negligible and residence time of the water column was high, there was minimal introduction of terrestrial NO3- into the streams and in-stream processing of NO3- was more apparent than during the wet periods. In addition to mineralisation and nitrification, volatilisation and assimilation by plant and algae was highly likely to occur in the stream further reducing the NO3- concentration and further fractionating the isotopic signature of NO3-.

Page 7 Line 21: I think the authors are referring to Table 2 here. This will be corrected

Page 8 Line 15 and 32: I strongly suggest the authors cite and discuss the implications of the outcomes from the work of Granger and Wankel, 2016 (Isotopic overprinting of nitrification on denitrification as a ubiquitous and unifying feature of environmental nitrogen cycling; PNAS) and how this may influence your interpretation of N turnover in your catchment. The study by Granger and Wankel (2016) will be discussed in the revised manuscript as suggested by the reviewer. Page 10 Line 5: It is worth noting that although the dual isotopic composition of $\delta$18O-NO3- and $\delta$15N-NO3- deviates from a trajectory of 1 (trajectory of 1 indicates denitrification), it is still a salient trend indicating the occurrence of denitrification and is consistent with the $\delta$18O-NO3-:$\delta$15N-NO3- recurrently observed in freshwater systems (Kendall et al. 2007). This deviation in our study could be explained by concurrent NO3- production catalysed by nitrification and/or annamox (Granger and Wankel 2016) although the significance of annamox is still disputable. Based on the multi-process model developed by Granger and Wankel (2016), the two most important factors in the nitrification pathway that govern the $\delta$18O of the newly produced NO3- are $\delta$18O of the ambient water and the flux of NO2- oxidation (Granger and Wankel 2016). Deflation of $\delta$18O-NO3-:$\delta$15N:NO3- trajectory below 1 observed in this study was likely to be associated with the low $\delta$18O-H2O values which contributed to lower $\delta$18O values for nitrified NO3-. Higher NO3- reduction rate versus NO2- oxidation rate which contributed to the $\delta$15N-enriched pool of nitrified NO3-, greater than the denitrified NO3- also drives the $\delta$18O-NO3-:$\delta$15N-NO3- trajectory to values below 1 (see Granger and Wankel 2016 for explanation). All in all, this highlights the significant contribution of nitrification along with denitrification in the WP catchment. Page 11 Line 5: NO3- in group B has variable $\delta$15N and $\delta$18O values as shown by Bunyip and Toomuc. This could be attributed to isotopic fractionation either during plant and/or algae uptake or denitrification as substantiated by the parallel increase of $\delta18O-NO3-$ versus $\delta15N-NO3-$ (Fig. 9). Based on Fig. 9, the large uncertainties in the $\delta18O-NO3-$ of the nitrified end members have resulted in overlapping of isotopic signatures of the three major sources (nitrified cow manure, nitrified inorganic fertiliser and nitrified SOM). All three sources appeared to have influenced the $\delta15N$ and $\delta18O$ of the residual NO3- in the stream. This scenario reinstates the sensitivity and the importance of accurately determining the $\delta18O-NO3-$ of the initial NO3- in the effort to apportion the relative contribution of different sources.

Page 10 section (3): an earlier introduction of the different behaviors of N and O isotopes during internal processing of nitrate will make this section easier to understand for the none expert reader. I would not put denitrification under the heading recycling, if the authors are referring to nitrate reduction, followed by reoxidation please say so. We will include a paragraph in the introduction to briefly discuss about the different behaviours of nitrate isotopes during internal processing of nitrate. We have also changed the term 'recycling' to 'internal processes'. Page 2 Line 7: To date, the most promising tool to investigate the sources and sinks of NO3- are the dual isotopic compositions of NO3- at natural abundance level (expressed as $\delta15N-NO3-$ and $\delta18O-NO3-$ in ‰. Preferential utilisation of lighter isotopes (14N and 16O) over heavier isotopes (15N and 18O) leads to distinctive isotopic signatures that differentiate the various NO3- sources/end members (e.g. inorganic and organic fertiliser, animal manure, atmospheric deposition) and the predictable kinetic fractionation effect when NO3- undergoes different biological processes (e.g. nitrogen fixation and denitrification). For instance, denitrification and phytoplankton assimilation fractionate N and O isotopes in a 1:1 pattern. Simultaneous measurement of $\delta15N-NO3-$ and $\delta18O-NO3-$ also provides complementary information on the cycling of NO3- in the environment. $\delta18O-NO3-$ is a more effective proxy of internal cycling of NO3- (i.e. assimilation, mineralisation and nitrification) compared to $\delta15N-NO3-$. This is because during NO3- assimilation and mineralisation, N atoms are recycled between fixed N pools and the O atoms are removed and replaced by nitrification (Sigman et al. 2009; Buchwald et

al. 2012).

Page 10 Line 11: Do the authors know when fertilizer is applied in this catchment, how does this align with your runoff / turnover hypotheses? Unfortunately we do not have the information on when fertiliser was applied in the catchment hence no further conclusion could be drawn on the relationship of fertiliser application and the runoff processes.

Figure 1: Please mark on the map of Australia where southern Victoria is. Figure 3: Mark on upper / lower Bunyip. Figure 5: Where have the authors taken these isotope effects from? Please cite the relevant literature in the caption. A positive / inverse isotope effect for nitrification? Figure 7: More details are needed in the figure caption, what do the crosses and dashed line mean? I also assume that it is the y intercept values determined in Figure 6 that have been plotted. Figure 9: define what starting values you have used and where they have come from, particularly for the oxygen isotopes. All the figures will be revised according to the reviewer's comments.

[Figure]

---

## Author Response (AR1)

**Stable isotopes of nitrate reveal different nitrogen processing mechanisms in streams across a land use gradient during wet and dry periods**

Response to Reviewer Comments

We thank the reviewers and the associate editor for their constructive comments. We have addressed the reviewers' comments individually (as detailed below) and have revised the manuscript accordingly. Please note that page/line numbers in reviewers' comments refer to the original manuscript while our references to page/line numbers refer to the revised manuscript.

**Reviewer #2**

GENERAL COMMENTS

Wong and co-authors present a study of stable isotopes of nitrate from five streams within the same catchment area in Southeast Australia sampled during wet and dry periods. The five streams show different degrees of land use intensities. The aim is to reveal different sources and transformation processes of nitrate compared between rainfall patterns through the isotopic composition (d15N-NO3- and d18O-NO3-). Results show that differences between wet and dry periods can be explained by the dominance of different sources on the isotopic composition. During wet periods artificial fertilizer was probably the main source, whereas nitrified organic matter in sediment and nitrified manure dominated the sources during dry periods. The manuscript is well written and presents the results in a logical order. The figures illustrate the findings very well. This novel dataset is suitable for publication in Biogeosciences, however, there are some points that should be addressed by the authors.

MAJOR COMMENTS

(1)  To study the impact of rainfall on isotopic composition in a more rigorous way, it would have been interesting to use samplings with different amounts of rainfall in the previous days (instead of only differentiating between wet and dry periods) in order to see whether rainfall and isotopic composition could be correlated. At least the authors should explain why such a study was not carried out.

There was no correlation between the isotope values (both $\delta^{15}N$ and $\delta^{18}O$) of the streams and the total amount of rainfall for 10 days before each sampling event as shown in Fig. 1. A linear relationship between streams nitrate isotopes and rainfall amount might be expected for a pristine environment, however for heavily anthropogenic-affected environments like the Western Port catchment; it is impossible to observe such correlation because of the dominance of other sources of nitrate. In this study, the condition of the soil (i.e. wet versus dry) in the catchment and the residence time of the rivers were affecting the occurrence and the extent of certain biological processes in the catchment and thus the isotope values of the residual nitrate rather than rainfall. We believe this has been shown nicely in the manuscript. The amount of rainfall in this study was used as a direct indicator of soil condition and residence time of the rivers, hence why the isotope data was grouped into wet and dry periods instead of the actual amount of rainfall. Fig. 1, however; is a strong evidence to show that the amount of rainfall had minimal effect on the isotope values in this study; supporting our contention on the insignificant direct contribution of rainfall amount to the overall nitrate dynamic in the catchment. Fig. 1 has now been included in the supplementary information in the revised manuscript.

[Figure]

Figure 1: Relationship between (a) $\delta^{15}N$-$NO_3^-$; (b) $\delta^{18}O$-$NO_3^-$ of the streams and the total amount of rainfall for 10 days before the sampling event.

(2) Could the data not have been explored more thoroughly, e.g. other statistical methods than linear regression in order to identify multiple sources? What about isotope mixing and emission modeling for source identification?

Apportioning the contribution of multiple sources of nitrate using the suggested models requires well-defined isotope values of the end members, accurate fractionation factors of the possible processes (i.e. denitrification and mineralisation) as well as the loads and rates of each possible source and process particularly for an emission model. Unfortunately, determining the fractionation factors of the processes was beyond the scope of this study. These models were not suitable in this study because there was significant overlapping of the end member nitrate isotope values as well as the lack of information on the rates of different types of fertiliser applications. Hence, only a qualitative assessment of the sources was presented in this study.

(3) Significant correlations with very low r2 for isotopic composition and % agriculture (Fig 4) are used as argument for "dominance of anthropogenic nitrogen inputs within the catchment". The discussion should include a more detailed comparison to studies which found a similar but much stronger correlation between d15N of nitrate and land use.

We have included more detailed comparisons to studies with similar findings in the revised manuscript.

Page 8 Line 11: Other researchers have also documented similar trends of enriched $\delta^{15}N$-$NO_3^-$ with increasing percentage agriculture. For example Harrington et al. 1998, Mayer et al. 2002 and Voss et al. 2006 observed highly significant positive relationships between percentage agriculture land area and $\delta^{15}N$-$NO_3^-$ with $r^2 \sim 0.7$. However, these studies showed comparatively narrower and more depleted ranges of $\delta^{15}N$-$NO_3^-$ with 2.0 to 7.3‰; 4 to 8‰ and -0.1 to 8.3‰; respectively, suggesting more subtle changes in $\delta^{15}N$-$NO_3^-$ over a large span of agriculture land areas in these studies compared to our study.

(4) Is there any information to take away from individual samplings within the same stream? There is no information on river flow rates, for example. Could patterns of isotope data within the streams be explained by mixing of sources or in-stream processing?

Individual samplings within the same stream did not provide sufficient data points for the interpretation on the processes governing the isotope values of the residual nitrate in the streams. For example, at Watson creek only three samples were obtained during each sampling trip. We are not confident to deduce any findings based on that even though the isotope biplot or the keeling plots for some of the sampling events showed significant correlations.

SPECIFIC COMMENTS

Page 2 line 11 : Kendall 2007 ; "et al." is missing
Reference has been corrected.

Page 2 line 21-24: please explain in more detail in which way rainfall patterns are different in the southern hemisphere compared to the northern hemisphere.
More detailed explanation has been added to the revised manuscript.
    Page 2 Line 30: The southern hemisphere tends to have more sporadic and variable rainfall patterns compared to the northern hemisphere and Australia is an example of this.

Page 2 line 28: please delete space in "samplin g".
Format has been corrected.

Page 3 line 21-23: what are the criteria to give the amount of rainfall for 5 days (5-10 days) before sampling of wet periods (dry periods)? Following up, what is the residence time of water in the aquifer and in the river?
There were no specific criteria used to classify the wet and dry periods in this study other than the amount of rainfall prior to the sampling dates. The samplings for wet periods were carried out after a few days of continuous rain. The number of days (5 for wet and 5-10 for dry) were solely to give the readers an idea on the duration of the rain. The same explanation applies for the dry period - the area had received no rain for 5 to 10 days. We also cross-checked the rainfall amount with the discharge of a few streams which were gauged in the area. The discharge of the streams was doubled during the wet periods compared to the dry periods. This information; however, was not discussed in the manuscript as we do not have the complete stream discharge dataset for all the studied streams. We also did not have the information for the residence time of water in both river and aquifer; hence rainfall was used as the qualitative indicator of the residence time of the river.

Page 4 line 10: a figure with some additional water quality parameters would be nice to include as a supplementary material.
This is a good suggestion but all the water quality parameters were relatively consistent throughout the sampling sites and there were no interesting trends or patterns across different sampling sites/periods. All the important data has been presented in the results section.

Page 4 line 16-18: how many samples of fertilizer and cow manure were analysed? Please specify.
A total of 4 fertiliser and 5 cow manure samples were analysed. This has now been specified in the revised manuscript.
    Page 4 Line 28: In addition to stream water and sediment, we also collected four samples of artificial/inorganic fertiliser (from the fertiliser distributor in the area) and five cow manure samples from local farmers.

Page 6 line 14/15: what about atmospheric deposition? It is only mentioned on page 7 line 1. Couldn't mixing lead to a depletion of the d18O, with NO3- from atmospheric deposition still contributing partly to the signal?

Contribution from atmospheric deposition although possible was not significant and this has been explained in the manuscript (Page Line). Mixing with atmospheric-$NO_3^-$ could potentially change the $\delta^{18}O$ of the residual $NO_3^-$ but it would only get more enriched rather than more depleted because $\delta^{18}O$ of atmospheric nitrate has been reported to be >60‰ in the literature.

Page 6 line 27: delete the d of "comprised".

This has been corrected.

Page 7 line 8-10: there is no statistic evidence given by the authors to show that there is an actual trend, so this should be rephrased.

These lines have been rephrased.

Page 8 Line 2: Agricultural land use (i.e. market gardens and cattle rearing) appeared to influence $NO_3^-$ concentrations in our study sites. As shown in Fig. 4(a), during the wet periods, high $NO_3^-$ concentrations (> 40 µM) were particularly observed at sites with more than 70% agricultural land use. During the dry periods, although $NO_3^-$ concentrations were generally lower than 36µM, the outliers were observed at sites with more than 70% agricultural land use.

Page 7 line 10-12: for the dry periods there are at least 6 data points with NO3- conc. > 50 µM, so "consistently lower" (than 36 µM) is not correct.

The word 'consistently' has been replaced by 'generally'.

Page 7 line 13: replace "entered" by "entering".

This has been corrected.

Page 7 line 21/22: please state clearly, that although significant, correlation coefficients r2 are 0.2 and 0.39, respectively, so quite low. From there on it is obvious that the relationship between d15N and % agriculture is not evident at all from this study. This has to be expressed more clearly.

We agree that the correlation between $\delta^{15}N$-$NO_3^-$ and percentage agriculture during the dry periods was low and could be marginally significant. However, we are convinced that there was a significant and strong relationship between the two variables during the wet periods. The low $r^2$ in Fig. 4 was due to 4 data points at Toomuc Creek which have skewed the relationship (as indicated in Fig. 2 in this document). The $r^2$ value increased to 0.58 (p value <0.01) when the 4 data points were excluded. We have rephrased all the related texts in the manuscript to reflect on this.

Page 8 Line 5: Similarly, enriched $\delta^{15}N$-$NO_3^-$ in the streams were mainly found at sites with high percentage agricultural land use (between 75 to 85%) for both dry and wet periods suggesting that enriched $\delta^{15}N$-$NO_3^-$ in the stream were originated from agricultural activities. In fact, the most enriched $\delta^{15}N$-$NO_3^-$ values (>30‰) were observed at the most downstream site of Watson Creek which has the largest percentage of market gardens (although the total agricultural area is not the highest amongst all the studied sites). We also observed a significant positive relationship between $\delta^{15}N$-$NO_3^-$ and percentage agriculture during the wet periods (Fig. 4b). This further supports the contention that agricultural activities were the main control of the $\delta^{15}N$-$NO_3^-$ in the streams.

[Figure]

Figure 2: Relationship between $\delta^{15}N$-$NO_3^-$ and percentage agriculture during the wet periods

Page 7 line 24-26: the comparison to the other studies has to be made more in detail. For example, Voss et al (2006) observed a significant correlation for 11 streams and weighted monthly means of d15N and % agriculture. If comparable to this data, it should be Figure 7 from this study (if it is correct that it represents average values per stream). And for this representation, there is no significant correlation.

The $\delta^{15}N$-$NO_3^-$ values in Figure 7 are not average values but they are the y-intercepts from the Keeling plot for individual stream (Figure 6). These values represent the predicted $\delta^{15}N$-$NO_3^-$ of the initial end member. As such, Figure 7 in this study is not comparable to the figure in the study by Voss et al. (2006). This has been made clearer in the figure caption.

Figure 7: Relationship between $\delta^{15}N$-$NO_3^-$ of the dominant initial source (indicated by the y-intercept of the Keeling plots; Figure 6) and percentage agriculture during wet periods. Data for Bass-dry period was also presented because only the Keeling plot for Bass-dry period indicates mixing between different sources. The shaded area represents the $\delta^{15}N$-TN of the potential end members.

Page 8 line 18-20 (and following sentences): Give first all the evidence that allows you to conclude that in-stream processing was not the dominant process for regulating the isotopic composition. These arguments could be supported by a more detailed discussion of the relevant literature.

This paragraph has been restructured as suggested by the reviewer.

Page 9 Line 12: In-stream processing of $NO_3^-$ was not evident during the wet periods based on the lack of relationships between $\delta^{18}O$-$NO_3^-$ and $[NO_3^-]$ as well as between $\delta^{18}O$-$NO_3^-$ and $\delta^{15}N$-$NO_3^-$ for the individual streams (shown in Supplementary Fig. 1). If denitrification was dominant, both $\delta^{15}N$-$NO_3^-$ and $\delta^{18}O$-$NO_3^-$ values are expected to increase at low $NO_3^-$ concentration and there would be systematic increase of both N and O isotopes of $NO_3^-$ (Fry 2008). In addition, high DO in the water column ruled out the possibility of pelagic denitrification.

Careful examination of the Keeling plots for individual streams (Fig. 6) revealed that during the wet periods, $NO_3^-$ concentrations were significantly and linearly correlated with $1/[NO_3^-]$ in all the streams. These relationships strongly suggest mixing between two sources (with distinctive isotopic signatures) as the dominant process regulating the isotopic composition of the residual $NO_3^-$ in the streams during the wet periods.

Page 8 line 20-22: Put figures of d18O vs [NO3-] and d18O vs d15N for individual streams in supplementary material to support your argument.

The relationships between $\delta^{18}O$ vs [NO$_3^-$] and $\delta^{18}O$ vs. $\delta^{15}N$ have been added as Figure S1 in supplementary material.

[Figure]

Figure S1: Relationships between (a) $\delta^{18}O$-NO$_3^-$ and $\delta^{15}N$-NO$_3^-$; (b) $\delta^{18}O$-NO$_3^-$ and NO$_3^-$ concentration

Page 9 line 23: add "be" in between "subsequently" and "nitrified".
This has been corrected.

Page 10 line 19-21: as stated above, according to figure 4 there is no significant correlation for [NO3-] and percentage agriculture and r2 for the corr. between d15N and percentage agriculture are low, so please rephrase this conclusion. Similarly please rephrase the related sentence in the abstract (page 1 line 17/18).
This has been explained in the earlier comment.

Figures
Fig. 1: indicate percentage agriculture for each sampling site.
Percentage agriculture for each of the sampling site has been added to Fig. 1 as suggested by the reviewer

[Figure]

Figure 1: Map of Western Port Bay (WPB) in southern Victoria, Australia and major rivers discharging into WPB. Closed circles represent sampling sites where surface water samples were obtained. Values in parentheses represent the % agriculture area.

Fig. 2: For the Watsons river the "distance from WPB" does not correspond to the values from Fig. 3 (max= 30 km).
The "distance from WPB" has been corrected.

[Figure]

Fig. 4: use A and B for the two panels. In the lower panel indicate which trend curve corresponds to which dataset.

The caption of the figure has been updated according to reviewer's comment.

Figure 4: Relationship between (a) $NO_3^-$ concentration; (b) $\delta^{15}N$-$NO_3^-$ and the percentage of agricultural land use. In (b) solid line represents the relationship between the variables during dry periods; dotted line represents wet periods.

Fig. 7: Are these average values per site? If so, please indicate the SD. For Bass river (dry period) the value is somewhat high compared to Fig. 6. Please explain.

These are not average values per site but the y-intercept value from the Keeling plots for individual stream. This has been made clearer in the caption of the figure.

Figure 7: Relationship between $\delta^{15}N$-$NO_3^-$ of the dominant initial source (indicated by the y-intercept of the Keeling plots; Figure 6) and percentage agriculture during wet periods. Data for Bass-dry period was also presented because only the Keeling plot for Bass-dry period indicates mixing between different sources. The shaded area represents the $\delta^{15}N$-TN of the potential end members.

**Reviewer #3**

GENERAL COMMENTS

In the manuscript Stable isotopes of nitrate reveal different nitrogen processing mechanisms in streams across a land use gradient during wet and dry periods, Wong et al present natural abundance nitrate isotopes from five streams across a land use gradient during wet and dry seasons, allowing them to elucidate the controls on sources and transformations of nitrate. This is an interesting dataset and the authors have been resourceful and knowledgeable in their presentation and interpretation of the data. However, prior to publication the manuscript would benefit from a clear and concise definition of terms, and a clearer explanation of the isotope effects and there subsequent implications, as currently it seems hard to follow in places for the none expert reader. An important aspect of the interpretation of this dataset is that there is a tight coupling of mineralisation and nitrification, resulting in no isotope effect being expressed and hence the 15N of organic matter / ammonium and nitrate are similar. Currently this is not fully explained until Page 9 Line 24, making it difficult to understand the authors interpretation of the data prior to this, explaining this earlier on in the discussion will enable the reader to follow your thoughts / interpretation. A good example of this is Page 6 Line 15/16, break this thought down and explain to the reader here the tight coupling between mineralization and nitrification and hence no isotope effect being observed.

We have added a few sentences to explain the minimal isotope effect of the combined reaction of mineralisation and nitrification as suggested by the reviewer.

Page 6 Line 31: Nitrogen isotope of the $NO_3^-$ produced from these end members usually retains the signature of the $\delta^{15}N$-TN as a result of tight coupling between mineralisation (production of ammonium from organic matter) and nitrification (oxidation of ammonium to $NO_3^-$) as well as the minimal isotopic fractionation of both processes. It is well documented in the literature that in soil environment, mineralisation causes a small isotopic fractionation ($\pm 1\permil$; Kendall et al. 2007) to the produced $NH_4^+$. In agricultural areas where $NH_4^+$ is rapidly consumed or assimilated by crops, nitrification rate is usually low and would also exert a small isotopic fractionation to the produced $NO_3^-$.

SPECIFIC COMMENTS

Page 2 Line 10: it would be valuable here to state that you are talking about kinetic isotope effects and not equilibrium.

This has been made clearer in the revised manuscript

Page 2 Line 8: Preferential utilisation of lighter isotopes ($^{14}N$ and $^{16}O$) over heavier isotopes ($^{15}N$ and $^{18}O$) leads to distinctive isotopic signatures that differentiate the various $NO_3^-$ sources/end members (e.g. inorganic and organic fertiliser, animal manure, atmospheric deposition) and the predictable kinetic fractionation effect when NO3- undergoes different biological processes (e.g. nitrogen fixation and denitrification).

Page 2 Line 23 to 25: for the none expert, please explain why rainfall patterns are different in the southern hemisphere and its subsequent effects.

This has been explained in the revised manuscript.

Page 2 Line 30: The southern hemisphere tends to have more sporadic and variable rainfall patterns compared to the northern hemisphere and Australia is an example of this. The variable rainfall patterns can modulate different efficiencies of denitrification in soils and thus different fractionation effects to the residual $NO_3^-$ pool.

Page 2 Line 32 to 33: start preparing your reader here, why are denitrification and assimilation more prevalent in wet periods. Study area: throughout this section you refer to the gradient in land use across the catchment, a map of this would be a great addition to the manuscript (or could maybe be added to Figure 1).

Following sentences have been added to the revised manuscript to explain why denitrification can be more prevalent in wet periods. A land use map have been added as supplementary material (Figure S2).

Page 3 Line 7: In some studies (e.g. Riha et al. 2014; Kaushal et al. 2011), denitrification and assimilation by plants and algae have been reported to be more prominent during the dry seasons compared to the wet seasons but in other studies (e.g. Murdiyarso et al. 2010; Enanga et al. 2016) denitrification appeared to be more prevalent during the wet seasons as precipitation induces saturation of soils resulting in oxygen depletion and thereby low redox potentials that favour denitrification.

[Figure]

Figure S2: Land use map of Western Port catchment

Page 3 Line 32: how do the authors think using an integrated signal could of biased their interpretation of the results?
We think the integrated signal could potentially bias the interpretation of the results; however, the integrated signal was the best representation of the percentage agriculture area in the catchment.

Page 5 Lines 9 to 10: the term total nitrogen needs to be defined here, as it is important for the mineralization discussion later on. Would particulate organic nitrogen not be a more suitable term? Also, please add in that the values are relative to AIR and the precision of the measurements.
Particulate organic matter is a more suitable term however this was not specifically measured in our study. We used $\delta^{15}N$ of total nitrogen of the soil to directly represent the soil organic portion as most of the nitrogen in soils is generally bound in organic forms. This has been explained more thoroughly in the revised manuscript.
        Page 6 Line 30: The average $\delta^{15}N$-TN value of soils is used to directly represent the soil organic portion as most of the nitrogen in soils is generally bound in organic forms.

Page 5 Lines 24: Please add in nitrite concentrations, to confirm for the reader that the values are less than 1% of the nitrate (as stated in the methods).
Nitrite concentrations have been added to the results section of the revised manuscript.
        Page 6 Line 6: Nitrite concentrations ranged between 0.1µmol/L and 0.4µmol/L.

Page 6 Line 3: Be clear that you are talking about 15N values here.
This sentence has been corrected to reflect more clearly on the $\delta^{15}N$ values.
        Page 6 Line 18: Overall, $\delta^{15}N$ of the riverine $NO_3^-$ spanned a wide range (+4 to +33‰).

Page 6 Line 4: The enriched 15N-nitrate values seem to be constrained to a thin band between 70 to 85% agriculture, but then drop away again at higher percentage agriculture, do the authors have any hypotheses for this?
We hypothesised that the drop off of $\delta^{15}N$-$NO_3^-$ at > 85% agriculture was due to recent and possibly over-fertilisation of $NH_4^+$ fertiliser resulting in active nitrification. As a large amount of $NH_4^+$ was available, oxidation of $NH_4^+$ to $NO_3^-$ became the rate-determining step resulted in large fractionation (-38‰ to -14‰; Casciotti et al. 2003) and depleted $\delta^{15}N$-$NO_3^-$ in the residual $NO_3^-$ pool. Unfortunately we could not test this hypothesis as we did not have the information on the rates of fertiliser application and nitrification. This would be a good avenue for future research.

Page 6 Line 5: Surely the same is true for the Bass.
We agree with the reviewer and Bass has been included as exhibiting the same effect as Bunyip in the revised manuscript.
        Page 6 Line 20: Among all sites, $\delta^{15}N$-$NO_3^-$ values in the Bunyip and Bass were relatively depleted (+4 to +12‰ for Bunyip and +10 to 12‰ for Bass), with the lower range found at upper Bunyip (+4 to +8‰).

Page 6 Line 16 to 28 and Equation 1: It would be valuable to explain to the readers the value of using both N and O isotopes i.e. N is recycled between fixed N pools and the O atoms are removed and then replaced by nitrification and thereby sensitive to internal processing (this

could come here or in the introduction). The authors need to discuss the more recent literature when introducing and determining the oxygen isotope signal imparted by nitrification, the work of Carly Buchwald is particularly pertinent here.

We have added a few sentences in the introduction to discuss the value of using both N and O isotopes. We have also discussed the calculation used to estimate $\delta^{18}O\text{-}NO_3^-$ imparted by nitrification in more detail. All the texts and figures in the manuscript have been revised to reflect on these changes.

Page 2 Line 7: To date, the most promising tool to investigate the sources and sinks of $NO_3^-$ are the dual isotopic compositions of $NO_3^-$ at natural abundance level (expressed as $\delta^{15}N\text{-}NO_3^-$ and $\delta^{18}O\text{-}NO_3^-$ in ‰). Preferential utilisation of lighter isotopes ($^{14}N$ and $^{16}O$) over heavier isotopes ($^{15}N$ and $^{18}O$) leads to distinctive isotopic signatures that differentiate the various $NO_3^-$ sources/end members (e.g. inorganic and organic fertiliser, animal manure, atmospheric deposition) and the predictable kinetic fractionation effect when $NO_3^-$ undergoes different biological processes (e.g. nitrogen fixation and denitrification). For instance, denitrification and phytoplankton assimilation fractionate N and O isotopes in a 1:1 pattern. Simultaneous measurement of $\delta^{15}N\text{-}NO_3^-$ and $\delta^{18}O\text{-}NO_3^-$ also provides complementary information on the cycling of $NO_3^-$ in the environment. $\delta^{18}O\text{-}NO_3^-$ is a more effective proxy of internal cycling of $NO_3^-$ (i.e. assimilation, mineralisation and nitrification) compared to $\delta^{15}N\text{-}NO_3^-$. This is because during $NO_3^-$ assimilation and mineralisation, N atoms are recycled between fixed N pools and the O atoms are removed and replaced by nitrification (Sigman et al. 2009; Buchwald et al. 2012).

Page 7 Line 6: The $\delta^{18}O$ of $NO_3^-$ generated by nitrification of these sources is decoupled from $\delta^{15}N\text{-}NO_3^-$ but relies on the oxygen isotope of water ($\delta^{18}O\text{-}H_2O$), oxygen isotope of dissolved oxygen ($\delta^{18}O\text{-}O_2$) as well as the kinetic and equilibrium isotope effects during the sequential oxidation of $NH_4^+$ to $NO_2^-$ then $NO_3^-$ (Casciotti et al. 2010; Buchwald et al. 2012). Previous culture studies (Casciotti et al. 2010; Buchwald and Casciotti 2010; Buchwald et al. 2012) and observations in various marine systems (Sigman et al. 2009; Granger et al. 2013; Rafter et al. 2013) have found that $\delta^{18}O$ values for nitrified $NO_3^-$ were within a few ‰ of the $\delta^{18}O\text{-}H_2O$. Hence, -5.3‰; the average value of $\delta^{18}O\text{-}H_2O$ is adopted to represent the lower estimate of $\delta^{18}O$ of the nitrified $NO_3^-$ in this study. In a system where equilibrium exchange of oxygen between $H_2O$ and $NO_2^-$ is negligible but respiration and denitrification are prevalent/co-occurring, $\delta^{18}O\text{-}NO_3^-$ can be much greater than that of $\delta^{18}O\text{-}H_2O$. In this study, the $\delta^{18}O\text{-}NO_3^-$ values were all more enriched than -5.3‰ suggesting the co-occurrence of a fractionating process, most likely denitrification (this is discussed in the following section). Based on this reason, using -5.3‰ can potentially underestimate the $\delta^{18}O$ of the nitrified $NO_3^-$. The conventional 2:1 ($\delta^{18}O\text{-}H_2O:\delta^{18}O\text{-}O_2$) fractional source contribution model (Equation 1) is therefore used to calculate the maximum estimate of $\delta^{18}O$ of the nitrified $NO_{3\text{-}}$ in our study which is +4.3‰ by using -5.3‰ for the average $\delta^{18}O\text{-}H_2O$ and +23.5‰ for $\delta^{18}O\text{-}O_2$.

Page 6 Line 29: Is this value for cow manure similar to the literature to date?
Yes these values are similar to the literature to date.

Page 7 Line 14: 'terrestrial' what are the authors referring to here, fertilizer, manure, leaf litter, please be consistent with the use of terms throughout.
This term refers to a combination of sources and this has been made clearer in the revised manuscript.

 …..in stream $NO_3^-$ comprised mainly of terrestrially derived $NO_3^-$ (i.e. inorganic fertiliser, manure and soil organic matter) entered the streams through surface runoff…

Page 7 Lines 15 onwards: a slight restructure here would be beneficial, you are presenting your conclusions before the evidence, discussing your isotope data first in this section would make it easier to follow.

This section has been restructured as suggested by the reviewer.

 Agricultural land use (i.e. market gardens and cattle rearing) appeared to influence $NO_3^-$ concentrations in our study sites. As shown in Fig. 4(a), during the wet periods, high $NO_3^-$ concentrations ($> 40$ µM) were particularly observed at sites with more than 70% agricultural land use. During the dry periods, although $NO_3^-$ concentrations were generally lower than 36µM, the outliers were observed at sites with more than 70% agricultural land use. Similarly, enriched $\delta^{15}N-NO_3^-$ in the streams were mainly found at sites with high percentage agricultural land use (between 75 to 85%) for both dry and wet periods suggesting that enriched $\delta^{15}N-NO_3^-$ in the stream were originated from agricultural activities. In fact, the most enriched $\delta^{15}N-NO_3^-$ values (>30‰) were observed at the most downstream site of Watson Creek which has the largest percentage of market gardens (although the total agricultural area is not the highest amongst all the studied sites). We also observed a significant positive relationship between $\delta^{15}N-NO_3^-$ and percentage agriculture during the wet periods (Fig. 4b) which further supports the contention that agricultural activities were the main control of the $\delta^{15}N-NO_3^-$ in the streams. Other researchers have also documented similar trends of enriched $\delta^{15}N-NO_3^-$ with increasing percentage agriculture. For example Harrington et al. 1998, Mayer et al. 2002 and Voss et al. 2006 observed highly significant positive relationships between percentage agriculture land area and $\delta^{15}N-NO_3^-$ with $r^2 \sim 0.7$. However, these studies showed comparatively narrower and more depleted ranges of $\delta^{15}N-NO_3^-$ with 2.0 to 7.3‰; 4 to 8‰ and -0.1 to 8.3‰; respectively, suggesting more subtle changes in $\delta^{15}N-NO_3^-$ over a large span of agriculture land areas in these studies compared to our study.

Given that none of the predicted sources of $NO_3^-$ in the Western Port catchment exhibited an initial $\delta^{15}N-NO_3^-$ of more than +6‰, the isotopically-enriched $NO_3^-$ as well as the variability of $NO_3^-$ concentrations observed in this study were consequences of a series of transformation processes. Hence, we propose the following factors to explain the heavy isotopes and the different $NO_3^-$ concentrations across different periods observed in our study:

(1) During the wet period when surface runoff was conspicuous and residence time of the water column was low, in-stream $NO_3^-$ comprised mainly of terrestrially derived $NO_3^-$ (i.e. fertilisers, manure and soil organic matter) and there was limited in-stream processing of these $NO_3^-$. The high $NO_3^-$ concentrations and the heavy $\delta^{15}N-NO_3^-$ values reflect the occurrence of mineralisation, nitrification and subsequent preferential denitrification of the isotopically lighter $NO_3^-$ source/s in either the waterlogged soil or in the soil zone underneath the market gardens before transport to the streams through surface runoff.

(2) During the dry periods when surface runoff was negligible and residence time of the water column was high, there was minimal introduction of terrestrial $NO_3^-$ into the streams and in-stream processing of $NO_3^-$ was more apparent than during the wet periods. In addition to mineralisation and nitrification, volatilisation and assimilation by plant and algae was highly likely to occur in the stream further reducing the $NO_3^-$ concentration and further fractionating the isotopic signature of $NO_3^-$.

Page 7 Line 21: I think the authors are referring to Table 2 here.
This has been corrected

Page 8 Line 15 and 32: I strongly suggest the authors cite and discuss the implications of the outcomes from the work of Granger and Wankel, 2016 (Isotopic overprinting of nitrification on denitrification as a ubiquitous and unifying feature of environmental nitrogen cycling; PNAS) and how this may influence your interpretation of N turnover in your catchment.
The study by Granger and Wankel (2016) has been discussed in the revised manuscript as suggested by the reviewer.

Page 10 Line 5: It is worth noting that although the dual isotopic composition of $\delta^{18}O$-$NO_3^-$ and $\delta^{15}N$-$NO_3^-$ deviates from a trajectory of 1 (trajectory of 1 indicates denitrification), it is still a salient trend indicating the occurrence of denitrification and is consistent with the $\delta^{18}O$-$NO_3^-$:$\delta^{15}N$-$NO_3^-$ recurrently observed in freshwater systems (Kendall et al. 2007). This deviation in our study could be explained by concurrent $NO_3^-$ production catalysed by nitrification and/or annamox (Granger and Wankel 2016) although the significance of annamox is still disputable. Based on the multi-process model developed by Granger and Wankel (2016), the two most important factors in the nitrification pathway that govern the $\delta^{18}O$ of the newly produced $NO_3^-$ are $\delta^{18}O$ of the ambient water and the flux of $NO_2^-$ oxidation (Granger and Wankel 2016). Deflation of $\delta^{18}O$-$NO_3^-$:$\delta^{15}N$:$NO_3^-$ trajectory below 1 observed in this study was likely to be associated with the low $\delta^{18}O$-$H_2O$ values which contributed to lower $\delta^{18}O$ values for nitrified $NO_3^-$. Higher $NO_3^-$ reduction rate versus $NO_2^-$ oxidation rate which contributed to the $\delta^{15}N$-enriched pool of nitrified $NO_3^-$, greater than the denitrified $NO_3^-$ also drives the $\delta^{18}O$-$NO_3^-$:$\delta^{15}N$-$NO_3^-$ trajectory to values below 1 (see Granger and Wankel 2016 for explanation). All in all, this highlights the significant contribution of nitrification along with denitrification in the WP catchment.

Page 11 Line 5: $NO_3^-$ in group B has variable $\delta^{15}N$ and $\delta^{18}O$ values as shown by Bunyip and Toomuc. This could be attributed to isotopic fractionation either during plant and/or algae uptake or denitrification as substantiated by the parallel increase of $\delta^{18}O$-$NO_3^-$ versus $\delta^{15}N$-$NO_3^-$ (Fig. 9). Based on Fig. 9, the large uncertainties in the $\delta^{18}O$-$NO_3^-$ of the nitrified end members have resulted in overlapping of isotopic signatures of the three major sources (nitrified cow manure, nitrified inorganic fertiliser and nitrified SOM). All three sources appeared to have influenced the $\delta^{15}N$ and $\delta^{18}O$ of the residual $NO_3^-$ in the stream. This scenario reinstates the sensitivity and the importance of accurately determining the $\delta^{18}O$-$NO_3^-$ of the initial $NO_3^-$ in the effort to apportion the relative contribution of different sources.

Page 10 section (3): an earlier introduction of the different behaviors of N and O isotopes during internal processing of nitrate will make this section easier to understand for the none expert reader. I would not put denitrification under the heading recycling, if the authors are referring to nitrate reduction, followed by reoxidation please say so.
We have included a paragraph in the introduction to briefly discuss about the different behaviours of nitrate isotopes during internal processing of nitrate. We have also changed the term 'recycling' to 'internal processes'.

Page 2 Line 7: To date, the most promising tool to investigate the sources and sinks of $NO_3^-$ are the dual isotopic compositions of $NO_3^-$ at natural abundance level (expressed as $\delta^{15}N$-$NO_3^-$ and $\delta^{18}O$-$NO_3^-$ in ‰). Preferential utilisation of lighter isotopes ($^{14}N$ and $^{16}O$) over heavier isotopes ($^{15}N$ and $^{18}O$) leads to distinctive isotopic signatures that differentiate the various $NO_3^-$

sources/end members (e.g. inorganic and organic fertiliser, animal manure, atmospheric deposition) and the predictable kinetic fractionation effect when $NO_3^-$ undergoes different biological processes (e.g. nitrogen fixation and denitrification). For instance, denitrification and phytoplankton assimilation fractionate N and O isotopes in a 1:1 pattern. Simultaneous measurement of $\delta^{15}N$-$NO_3^-$ and $\delta^{18}O$-$NO_3^-$ also provides complementary information on the cycling of $NO_3^-$ in the environment. $\delta^{18}O$-$NO_3^-$ is a more effective proxy of internal cycling of $NO_3^-$ (i.e. assimilation, mineralisation and nitrification) compared to $\delta^{15}N$-$NO_3^-$. This is because during $NO_3^-$ assimilation and mineralisation, N atoms are recycled between fixed N pools and the O atoms are removed and replaced by nitrification (Sigman et al. 2009; Buchwald et al. 2012).

Page 10 Line 11: Do the authors know when fertilizer is applied in this catchment, how does this align with your runoff / turnover hypotheses?

Unfortunately we do not have the information on when fertiliser was applied in the catchment hence no further conclusion could be drawn on the relationship of fertiliser application and the runoff processes.

Figure 1: Please mark on the map of Australia where southern Victoria is.

The red circle in Figure 1 indicates the location of southern Victoria

[Figure]

Figure 3: Mark on upper / lower Bunyip.

Figure 3 has been updated in the revised manuscript:

[Figure]

Figure 5: Where have the authors taken these isotope effects from? Please cite the relevant literature in the caption. A positive / inverse isotope effect for nitrification?

More details have been provided in the caption of Figure 5:

Figure 5: Conceptual diagram illustrating the sources and processes of $NO_3^-$ during the wet and dry periods in the Western Port catchment. The values of enrichment factor ($\varepsilon$) were obtained from the literature (Kendall et al. 2007) to indicate the relative contribution of the transformation processes to the isotopic compositions of the residual $NO_3^-$.

Figure 7: More details are needed in the figure caption, what do the crosses and dashed line mean? I also assume that it is the y intercept values determined in Figure 6 that have been plotted.

More details have been added to the caption of Figure 7:

Figure 7: Relationship between $\delta^{15}N$-$NO_3^-$ of the dominant initial source (indicated by the y-intercept of the Keeling plots; Figure 6) and percentage agriculture during wet periods. Data for Bass-dry period was also presented because only the Keeling plot for Bass-dry period indicates mixing between different sources. The shaded area represents the $\delta^{15}N$-TN of the potential end members.

Figure 9: define what starting values you have used and where they have come from, particularly for the oxygen isotopes.

More details have been added to the caption of Figure 9:

Figure 9: Biplot of $\delta^{15}N$-$NO_3^-$ versus $\delta^{18}O$-$NO_3^-$ for Bunyip and Toomuc (group B data in Fig. 8b). Shaded areas represent theoretical assimilation trends for cow manure, SOM and inorganic fertiliser. The minimum starting value for $\delta^{18}O$-$NO_3^-$ was estimated from the average $\delta^{18}O$-$H_2O$ and the maximum $\delta^{18}O$-$NO_3^-$ value was estimated from Equation 1. The starting $\delta^{15}N$-$NO_3^-$ is the $\delta^{15}
[revised manuscript text omitted]

---

## Author Response (AR3)

**Stable isotopes of nitrate reveal different nitrogen processing mechanisms in streams across a land use gradient during wet and dry periods**

**Response to Reviewer Comments**

We thank the reviewers and the associate editor for their constructive comments. We have addressed the comments by both reviewers (as detailed below) and have revised the manuscript accordingly. Please note that page/line numbers in the reviewer's comments refer to the original manuscript while our references to page/line numbers refer to the revised manuscript.

REVIEWER #2

The revised manuscript by Wong and colleagues is much improved; laying out their arguments more clearly for the none-expert reader and I thank the authors for taking the time to address both mine and the other reviewers concerns. I have a few relatively minor suggestions prior to publication, mainly asking for a number of newly added sentences to be clarified.

Section 2.1 Study area: please refer to Figure S2 throughout this section, to support your description.
The reference to Figure S2 has been added.
        Page 4 Line 4: The five streams were selected based on the extent and distribution of land use types between and within each stream sub-catchment (see Fig. S2 in supplementary material), thus enabling comparisons within and between the streams.

Page 5, Line 22: Please provide the precision / reproducibility of the elemental analyzer measurements.
The precision of the EA-IRMS has been added to the revised manuscript.
        Page 5 Line 22: The precision of the elemental analysis and $\delta^{15}N$ was 0.5µg and ±0.2‰ (n=5), respectively.

Page 7, Lines 3 to 6: 'minimal isotopic fractionation of both processes' is this really the case for nitrification. Based on the literature ammonia oxidation / nitrification can be associated with a large isotope effect (e.g. Mariotti et al, 1981; Casciotti et al, 2003 and Santoro and Casciotti, 2011). Surely it is the tight coupling of processes that is resulting in no isotope effect being expressed and hence the 15N of organic matter / ammonium and nitrate being similar.
We agree with the reviewer and have now removed all the related text on the minimal fractionation factor resulted by nitrification.

Page 7, Lines 12 to 15: Here the authors are discussing how nitrate produced from nitrification can be fractionated by subsequent processes such as denitrification and this is of course correct. However, I am confused how this fits into this section about calculating the 18O of newly produced nitrate from nitrification. Put your use of equation one in context, tight coupling of the steps of nitrification, little exchange or nitrite accumulation and why not

to use the equation presented by Buchwald and Casciotti, 2010. Most of these points the authors already highlight, but the text needs to be reformulated to make this clearer, and I would suggest to remove or clarify the discussion of denitrification / a subsequent fractionation process.

As suggested by the reviewer, the discussion on denitrification as a subsequent fractionation process has been removed to avoid confusion on the isotope effects imparted by nitrification. We have also replaced Equation 1 with the equation suggested by Buchwald et al. 2012. To calculate the maximum and minimum estimates of $\delta^{18}O$ of newly produced nitrate from nitrification ($\delta^{18}O\text{-}NO_3^-{}_{final}$) using Equation 1, two assumptions were made:

(1) Ammonia was fully oxidised to $NO_3^-$ as no accumulation of $NO_2^-$ was observed in our system, hence the fraction of nitrite oxygen atoms exchanged with $H_2O$ during nitrite oxidation ($x_{NO}$) was not taken into account

(2) Full exchange of oxygen isotopes between nitrite and $H_2O$ during ammonia oxidation because all the observed $\delta^{18}O\text{-}NO_3^-$ values were more enriched than the average $\delta^{18}O$ of $H_2O$, hence $x_{AO} = 1$ was used in the equation

The minimum estimate of $\delta^{18}O\text{-}NO_3^-{}_{final}$ was calculated using the lower range of $^{18}\varepsilon_k\text{-}O_2 + {}^{18}\varepsilon_k\text{-}H_2O,_1$ (17.9‰) and $^{18}\varepsilon_k\text{-}H_2O,_2$ (12.8‰) while the maximum estimate was calculated using the upper range of $^{18}\varepsilon_k\text{-}O_2 + {}^{18}\varepsilon_k\text{-}H_2O,_1$ (37.6‰) and $^{18}\varepsilon_k\text{-}H_2O,_2$ (18.2‰). All the related texts and figures in the original manuscript have been updated.

Equation 1:

$$\delta^{18}O_{NO_3^-,final} = \left[\frac{2}{3} + \frac{1}{3}x_{AO}\right]\delta^{18}O_{H_2O} + \frac{1}{3}\left[(\delta^{18}O_{O_2} - {}^{18}\varepsilon_{k,O_2} - {}^{18}\varepsilon_{k,H_2O,1})(1 - x_{AO}) - {}^{18}\varepsilon_{k,H_2O,2}\right] + \frac{2}{3}{}^{18}\varepsilon_{eq}x_{AO}$$

Page 7 Line 2: The $\delta^{18}O$ of $NO_3^-$ generated by nitrification of these sources ($\delta^{18}O\text{-}NO_3^-{}_{final}$) is, however; decoupled from $\delta^{15}N\text{-}NO_3^-$. As shown in Equation (1) – adapted from Buchwald et al.2012, $\delta^{18}O\text{-}NO_3^-{}_{final}$ relies on the oxygen isotope of water ($\delta^{18}O\text{-}H_2O$), oxygen isotope of dissolved oxygen ($\delta^{18}O\text{-}O_2$), the kinetic isotope fractionation associated with incorporation of oxygen during ammonia oxidation ($^{18}\varepsilon_k\text{-}O_2$), kinetic isotope fractionation associated with incorporation of oxygen from water during ammonia oxidation ($^{18}\varepsilon_k\text{-}H_2O,_1$) and nitrite oxidation ($^{18}\varepsilon_k\text{-}H_2O,_2$), equilibrium isotope effect associated with oxygen isotope exchange between nitrite and water ($^{18}\varepsilon_{eq}$) as well as the fraction of nitrite oxygen atoms exchanged with $H_2O$ during ammonia oxidation ($x_{AO}$) (Casciotti et al. 2010; Buchwald et al. 2012). To date, $^{18}\varepsilon_k\text{-}O_2 + {}^{18}\varepsilon_k\text{-}H_2O$ cannot be separated. Previous culture studies have reported the overall $^{18}\varepsilon_k\text{-}O_2 + {}^{18}\varepsilon_k\text{-}H_2O,_1$ to range between 17.9‰ to 37.6‰ (Casciotti et al. 2010) while $^{18}\varepsilon_k\text{-}H_2O,_2$ ranges from 12.8‰ to 18.2‰ (Buchwald and Casciotti 2010). These values together with $^{18}\varepsilon_{eq}$ value of 14‰, average $\delta^{18}O\text{-}H_2O$ of -5.3‰ and $\delta^{18}O\text{-}O_2$ of 23.5‰ were used to calculate the maximum and minimum estimates of the $\delta^{18}O$ of newly produced $NO_3^-$ from nitrification. The minimum estimate of $\delta^{18}O\text{-}NO_3^-{}_{final}$ was calculated using the lower range of $^{18}\varepsilon_k\text{-}O_2 + {}^{18}\varepsilon_k\text{-}H_2O,_1$ (17.9‰) and $^{18}\varepsilon_k\text{-}H_2O,_2$ (12.8‰) while the maximum estimate was calculated using the upper range of $^{18}\varepsilon_k\text{-}O_2 + {}^{18}\varepsilon_k\text{-}H_2O,_1$ (37.6‰) and $^{18}\varepsilon_k\text{-}H_2O,_2$ (18.2‰). Based on the assumptions that ammonia was fully oxidised to $NO_3^-$ (as no accumulation of $NO_2^-$ was observed during our study period) and there was complete exchange of oxygen isotope between nitrite and $H_2O$ during ammonia oxidation ($x_{AO}=1$); which likely characterizes most freshwater systems (Casciotti et al. 2007, Snider et al. 2010,

Buchwald and Casciotti 2013); we calculated the $\delta^{18}O$ of produced $NO_3^-$ from nitrification to be between -2.03‰ and -0.23‰.

[Figure]

Figure 8: Biplot of $\delta^{15}N\text{-}NO_3^-$ versus $\delta^{18}O\text{-}NO_3^-$ for (a) wet and (b) dry periods. Blue shaded area represents possible isotopic compositions of denitrified $NO_3^-$ originated from SOM ($\delta^{15}N$: +4.5‰). Grey shaded area represents the possible isotopic composition of denitrified $NO_3^-$ originated from inorganic fertiliser ($\delta^{15}N\text{-}NO_3^-$: +0.1‰). The $\delta^{18}O\text{-}NO_3^-$ used were -2.3‰ and +-0.23‰ representing the minimum and maximum estimates of $\delta^{18}O$ of nitrified $NO_3^-$, respectively. The shaded area were plotted based on the theoretical 1:1 and 2:1 denitrification relationships between $\delta^{15}N\text{-}NO_3^-$ and $\delta^{18}O\text{-}NO_3^-$ (Kendall et al. 2007).

[Figure]

Figure 9: Biplot of $\delta^{15}N$-$NO_3^-$ versus $\delta^{18}O$-$NO_3^-$ for Bunyip and Toomuc (group B data in Fig. 8b). Shaded areas represent theoretical assimilation trends for cow manure, SOM and inorganic fertiliser. The maximum and minimum starting values for $\delta^{18}O$-$NO_3$- were estimated from Equation 1. The starting $\delta^{15}N$-$NO_3^-$ is the $\delta^{15}N$-TN value of respective end member. Solid and dotted lines represent the assimilation trends for Bunyip (both lower and upper Bunyip) and Toomuc, respectively. Assimilation rather than denitrification was considered a more plausible process controlling the distribution pattern for the group B dataset as the water column was oxic throughout the study period.

Page 10, Lines 5 to 16
- Until now in the manuscript you have discussed that denitrification results in a 2:1 pattern. Now you have switched to phrases such as 'trajectory of 1', put this in context for the reader, 2:1 versus 1:1 and how Granger and Wankel, 2016 are trying to reconcile this.
- 'anammox is still disputable' what are the authors referring to here, that anammox has not been observed in your system?
We have now standardised the use of the term.

   Page 10 Line 5: In fact, the deviation of the $\delta^{18}O$-$NO_3^-$:$\delta^{15}N$-$NO_3^-$ from the 1:1 trend to 2:1 corroborates the co-existence of other processes in our system (i.e. nitrification and/or anammox) in addition to denitrification. Based on the multi process model developed by Granger and Wankel (2016), the negative deflation of the denitrification trend (1:1) is strongly driven by concurrent $NO_3^-$ production catalysed by nitrification and/or anammox (Granger and Wankel 2016) when the rate of $NO_3^-$ reduction to $NO_2^-$ (via denitrification) is higher than the rate of $NO_2^-$ oxidation to $NO_3^-$ (via nitrification and/or anammox). Higher reduction rate of $NO_3^-$ to $NO_2^-$ tends to create a $NO_2^-$ pool with enriched $\delta^{15}N$ due to isotopic fractionation (0‰ to 20‰) during the reduction of $NO_2^-$ to $N_2$ (the last step of denitrification). The subsequent oxidation of the $\delta^{15}N$-enriched $NO_2^-$ leads to the production of $NO_3^-$ which is isotopically more enriched than denitrified $NO_3^-$ owing to inverse kinetic fractionation effects (-35‰ to 0‰); driving the negative deviation of $\delta^{18}O$-$NO_3^-$:$\delta^{15}N$-$NO_3^-$ from the 1:1 trend (Granger and Wankel 2016). During the wet periods, simultaneous occurrence of these three processes (nitrification, annamox and denitrification) was plausible due to the redox dynamics in the waterlogged soil zone. Downward percolation of oxygenated rain water could induce nitrification while denitrification and anammox could be promoted in the anoxic interstitial spaces of the waterlogged soil zone.

Page 11, Lines 5 to 7 and Figure 9: Here you are also using a 1:1 line and this needs to be explained for the reader and also highlighted in the caption of Figure 9 (the whole caption of Figure needs to be looked at, as currently information is missing, it is correct in the reviewer response). Surely you can also exclude denitrification here, due to the high levels of DO?
The caption of Figure 9 has been updated and the related texts have also been corrected.

Page 11 Line 6: $NO_3^-$ in group B has variable $\delta^{15}N$ and $\delta^{18}O$ values as shown by Bunyip and Toomuc. This could be attributed to isotopic fractionation during plant and/or algae uptake of $NO_3^-$ as substantiated by the parallel increase of $\delta^{18}O$-$NO_3^-$ versus $\delta^{15}N$-$NO_3^-$ (Fig. 9). Denitrification was ruled out due to high levels of dissolved oxygen in the water column. Close convergence of the linear relationships onto the theoretical assimilation trends of the nitrified artificial fertiliser and SOM (Fig. 9) reiterate the dominant contribution of these sources to the riverine $NO_3^-$ during the dry periods. It is worth noting that the initial $\delta^{18}O$ of nitrified $NO_3^-$ was estimated assuming full O isotopic equilibration between $NO_2^-$ and $H_2O$. Partial O isotope disequilibrium which was possible could affect the initial $\delta^{18}O$ signature of nitrified $NO_3^-$. If this happened, the minimum estimate of $\delta^{18}O$ of nitrified $NO_3^-$ could be more depleted and the overall linear relationship of $\delta^{18}O$-$NO_3^-$:$\delta^{15}N$-$NO_3^-$ would shift, resulting in more obvious contribution of artificial fertiliser, SOM and possibly organic fertiliser (Fig, 9). This scenario emphasizes the sensitivity of the initial $\delta^{18}O$ of nitrified $NO_3^-$ in determining the relative contribution of multiple sources in the catchment.

Figure 9: Biplot of $\delta^{15}N$-$NO_3^-$ versus $\delta^{18}O$-$NO_3^-$ for Bunyip and Toomuc (group B data in Fig. 8b). Shaded areas represent theoretical assimilation trends for cow manure, SOM and inorganic fertiliser. The maximum and minimum starting values for $\delta^{18}O$-$NO_3$- were estimated from Equation 1. The starting $\delta^{15}N$-$NO_3^-$ is the $\delta^{15}N$-TN value of respective end member. Solid lines represent the assimilation trends for Bunyip (both lower and upper Bunyip) and Toomuc. Assimilation rather than denitrification was considered a more plausible process controlling the distribution pattern for the group B dataset as the water column was oxic throughout the study period.

Page 11, Lines 22 to 24: Should this be Figure 9? Otherwise I do not understand this sentence as currently written.
Yes this was supposed to be Fig. 9. This has now been corrected in the revised manuscript.

Figure 5: Reference is missing from caption, but is present in the reviewer response, please make sure it is in the final version. I missed this in my last review, but I can see that in this figure you are showing negative isotope effects, whereas in the text they are positive (e.g. volatilization, Page 7, Line 23). I know the use of positive versus negative varies in the literature, but please be consistent with your use throughout the manuscript.
The caption of Figure 5 and the enrichment factors have been updated to align with the texts in the manuscript.

Figure 5: Conceptual diagram illustrating the sources and processes of $NO_3^-$ during the wet and dry periods in the Western Port catchment. The values of enrichment factor ($\varepsilon$) were obtained from the literature (Kendall et al. 2007) to indicate the relative contribution of the transformation processes to the isotopic compositions of the residual $NO_3^-$.

[Figure]

Figure 5: Conceptual diagram illustrating the sources and processes of $NO_3^-$ during the wet and dry periods in the Western Port catchment. The values of enrichment factor ($\varepsilon$) were obtained from the literature (Kendall et al. 2007) to indicate the relative contribution of the transformation processes to the isotopic compositions of the residual $NO_3^-$.

REVIEWER #3

On page 2 line 32 the authors should give some references
Two references have been added to the revised manuscript.
        Page 2 Line 32: The variable rainfall patterns can modulate different efficiencies of denitrification in soils and thus different fractionation effects to the residual $NO_3^-$ pool (Chien et al. 1977, Billy et al. 2010)

On page line 22 I suggest that the authors add figures of at least DO in the supplementary material
A figure showing the %DO saturation has been added as supplementary material
        Page 6 Line 2: The streams were oxic throughout the course of our study with %DO saturation between 60 to 110% (shown in Supplementary Fig. S2).
        Figure S2: Percent saturation of dissolved oxygen for all the sampling sites. Black dots represent dry periods and grey dots represent wet periods. 'S' represents sampling site.

[Figure]

Figure S2: Percent saturation of dissolved oxygen for all the sampling sites. Black dots represent dry periods and grey dots represent wet periods. 'S' represents sampling site.

On page 8 line the authors should give the r2 and p value of their correlation and also in the subsequent parts of discussion, conclusion and abstract where reference to the correlation between % agriculture and nitrate isotopes is made these values should always be mentioned. Although I agree with the authors that there is a higher coeff. of correlation when the data of one creek is excluded, I don't see any valid argument why this should be done. Therefore, I still think that the manuscript gives too much attention to a very weak correlation.

The $r^2$ and p values have been added to the revised manuscript as suggested by the reviewer. The main focus of the manuscript was to investigate the effects of different rainfall conditions on the dominant sources and transformation processes of nitrate in the streams using stable

isotopes. We don't agree that the manuscript has given too much attention to a very week correlation. The correlation between % agriculture and nitrate isotopes was presented mainly to indicate that agriculture was the major land use affecting the nitrate dynamic in the stream. As discussed earlier, the weak correlation was due to one stream and this would be expected especially when the degree of nitrate processing might be different between different streams.

Page 1 Line 18: At the catchment scale, we observed significant positive relationships between $NO_3^-$ concentrations ($p < 0.05$), $\delta^{15}N$-$NO_3^-$ ($p < 0.01$) and percentage agriculture (particularly during the wet period) reflecting the dominance of anthropogenic nitrogen inputs within the catchment.

Page 8 Line 9: We also observed a significant positive relationship between $\delta^{15}N$-$NO_3^-$ and percentage agriculture during the wet periods ($r^2 = 0.39$, $p<0.01$; Fig. 4b).

[revised manuscript text omitted]
 $\delta^{18}O\text{-}NO_3^-$ were estimated from Equation 1. The starting $\delta^{15}N\text{-}NO_3^-$ is the $\delta^{15}N\text{-}TN$ value of respective end member. Solid and dotted lines represent the assimilation trends for Bunyip (both lower and upper Bunyip) and Toomuc, respectively. Assimilation rather than denitrification was considered a more plausible process controlling the distribution pattern for the group B dataset as the water column was oxic throughout the study period.**